# Revisiting the Evaluation of Image Synthesis with GANs

**Mengping Yang**[1,†]   **Ceyuan Yang**[1,†]   **Yichi Zhang**[1]   **Qingyan Bai**[3]   **Yujun Shen**[2]   **Bo Dai**[1]

[1]Shanghai AI Laboratory   [2]Ant Group   [3]Tsinghua University

## Abstract

A good metric, which promises a reliable comparison between solutions, is essential for any well-defined task. Unlike most vision tasks that have per-sample ground-truth, image synthesis tasks target generating *unseen* data and hence are usually evaluated through a distributional distance between one set of real samples and another set of generated samples. This study presents an empirical investigation into the evaluation of synthesis performance, with generative adversarial networks (GANs) as a representative of generative models. In particular, we make in-depth analyses of various factors, including how to represent a data point in the representation space, how to calculate a fair distance using selected samples, and how many instances to use from each set. Extensive experiments conducted on multiple datasets and settings reveal several important findings. Firstly, a group of models that include both CNN-based and ViT-based architectures serve as reliable and robust feature extractors for measurement evaluation. Secondly, Centered Kernel Alignment (CKA) provides a better comparison across various extractors and hierarchical layers in one model. Finally, CKA is more sample-efficient and enjoys better agreement with human judgment in characterizing the similarity between two internal data correlations. These findings contribute to the development of a new measurement system, which enables a consistent and reliable re-evaluation of current state-of-the-art generative models. [1]

## 1   Introduction

Through reproducing realistic data distribution, generative models [67, 2, 19, 38, 28, 52] have enabled thrilling opportunities to go beyond existing observations via content recreation. Their technical breakthroughs in recent years also directly lead to the blooming of metaverse, AI Generated Content (AIGC), and various other downstream applications. However, accurately measuring the progress and performance of generative models poses significant challenges, as it requires a consistent and comprehensive evaluation of the divergence between real and synthesized data distributions. Among existing evaluation metrics [41, 4, 34, 44], Fréchet Inception Distance (FID) [22] is the most popular evaluation paradigm for synthesis comparison. Despite its popularity, recent studies [35, 44, 36, 1, 26] have identified several flaws in the FID metric that may miscalculate the actual improvements of generative models. Consequently, there is a pressing need for a more systematic and thorough investigation in order to provide a more accurate assessment of synthesis performance.

Therefore, this paper presents an empirical study that rigorously revisits the consistency and comprehensiveness of evaluation paradigms for generative models. Commonly used paradigms including FID typically contain two key components: a feature extractor $\phi(\cdot)$ and a distributional distance $d(\cdot)$. Through $\phi(\cdot)$, *i.e.,* Inception-V3 [56], a number of real samples ($x \in X$) and synthesized

---

† denotes equal contribution.

[1]code is available at https://github.com/kobeshegu/Synthesis-Measurement-CKA.

ones ($y \in Y$) are projected into a pre-defined representation space to approximate the respective data distributions. $d(\cdot)$, *i.e.,* Fréchet Distance [17] is then calculated in the space to deliver the similarity index, indicating the synthesis quality. Following the philosophy, our study revolves around the representation space defined by the feature extractor $\phi(\cdot)$, and the distributional distance $d(\cdot)$.

The most commonly used feature extractor, *i.e.,* Inception-V3, has been found to encode limited semantics and possesses a large perceptual null representation space [35], making it hardly reflect the actual improvement of synthesis. Accordingly, we gather multiple models that vary in *supervision signals*, *architectures*, and *representation similarities* to investigate the impact of the feature extractor $\phi(\cdot)$, which are respectively motivated by 1) representation spaces defined by the extractor $\phi(\cdot)$ usually encode different levels of semantics, understanding which extractor or set of extractors can capture rich semantics is crucial yet less explored; 2) and it remains uncertain how the correlations between various representation spaces affect the evaluation results. In addition to studying the feature extractor $\phi(\cdot)$, we further delve into *the consistency across spaces*, *the choice of distances*, and *the number of evaluated samples* for the distributional distance $d(\cdot)$. It is imperative that the distributional distance consistently provides a reliable similarity index, even when measured in various representation spaces. Similarly, selecting an appropriate number of samples to represent the synthesis distribution is a critical consideration. Regarding the choice of $d(\cdot)$, besides Fréchet Distance, we incorporate Centered Kernel Alignment (CKA) [11, 10] to explore alternatives for a more accurate evaluation.

In order to qualitatively and quantitatively compare different choices of the aforementioned aspects of $\phi(\cdot)$ and $d(\cdot)$, we re-implement the visualization pipeline and the histogram matching technique as in [35]. These techniques allow us to respectively highlight the most relevant semantics of an image *w.r.t* the similarity indexes and to attack the measurement system through a selected subset. Moreover, we conduct an extensive user study involving 100 participants to investigate the correlation between the synthesis measurement and human judgment. Through these analysis tools, we make *several significant findings*: 1) One specific extractor (*e.g.,* Inception-V3) tends to capture limited semantics and provide unreliable measurement results. 2) Various extractors naturally focus on different aspects of semantics thus demonstrating the potential generalization across different domains, motivating us to incorporate multiple extractors to deliver a comprehensive and reliable measurement. 3) With respect to features obtained from multiple representation spaces defined by different extractors, CKA proves to be effective in measuring the discrepancy and produces bounded values that facilitate comparison across different spaces. 4) In conjunction with the extensive user study, CKA consistently agrees with human judgment whereas FID failed in some circumstances, further underscoring the advantages of using CKA as the evaluation metric.

After revisiting various factors, we leverage the newly developed measurement system to re-evaluate extensive generative models under various settings. Current state-of-the-art generative models on several domains are first benchmarked through our system. Moreover, the performances and intrinsic properties of diffusion models and GANs are comprehensively compared with the new measurement system. Furthermore, the measurement system is employed to evaluate the performance of image-to-image translation models. It turns out that our system not only delivers a similar assessment with FID and human evaluation in most cases, but also demonstrates a more reliable and consistent correlation with human judgment than FID, demonstrating the robustness and superiority of the proposed metric.

## 2 Preliminary

This section briefly introduces the feature extractor $\phi(\cdot)$, distributional distance $d(\cdot)$, evaluated datasets and generative models, as well as auxiliary analysis approaches used in our study.

### 2.1 Representation Spaces

To investigate the effect of feature extractors $\phi(\cdot)$, we gather multiple models that are pre-trained on different objectives in fully-supervised/self-supervised manners, and with various architectures (*e.g.,* ViT, CNN, and MLP).

**Supervision.** Models that are trained in fully-supervised and self-supervised manners are collected due to their potential for generalization. In particular, we include the backbone networks which are

well-trained on supervised ImageNet classification tasks [13, 21]. Further, we gather the weights derived from single-modal/multi-modal self-supervised learning approaches [46, 8, 9, 6].

**Architecture.** In addition to considering models trained with different supervisions, we also include models with various architectures. Concretely, models with CNNs [56, 53, 21, 6, 8], ViTs [9, 39, 46], as well as MLPs [61] architectures are gathered together for our investigation.

| Properties | Feature Extractors |
|---|---|
| **Supervision** | Fully-supervised [56, 21], Self-supervised [46, 8, 9, 6] |
| **Architecture** | CNNs [21, 46, 8], ViTs [39, 46, 9], MLPs [61, 59, 37] |

## 2.2 Distributional Distances

**Fréchet Inception Distance (FID)** computes the Fréchet Distance [17] between two estimated Gaussian distributions, *i.e.*, $\mathcal{N}(\mu_s, \Sigma_s)$ and $\mathcal{N}(\mu_r, \Sigma_r)$, which represent the feature distributions of synthesized and real images extracted by the pre-trained Inception-V3. Formally, Fréchet Distance (FD) is calculated by

$$\mathrm{FD}(\mathrm{X}, \mathrm{Y}) = \|\mu_s - \mu_r\|^2 + \mathrm{Tr}\left(\Sigma_s + \Sigma_r - 2\left(\Sigma_s \Sigma_r\right)^{\frac{1}{2}}\right), \quad (1)$$

where X and Y represent the real distribution and synthesized distribution, respectively. $\mu$ and $\Sigma$ correspond to the mean and variance of Gaussian distribution, and $\mathrm{Tr}(\cdot)$ is the trace operation.

**Centered Kernel Alignment (CKA)** as a widely used similarity index for quantifying neural network representations [11, 32, 12], could also serve as a metric of similarity between two given data distributions. CKA has been identified to have several advantages: 1) CKA is invariant to orthogonal transformation and isotropic scaling, making is stable under various image transformations; 2) CKA can capture the non-linear correspondence between representations benefit from its kernel mapping; and 3) CKA can determine the correspondence across different features and with different widths, whereas previous metrics fail [32].

Formally, CKA is normalized from Hilbert-Schmidt Independence Criterion (HSIC) [20] to ensure invariant to isotropic scaling and is calculated by

$$\mathrm{CKA}(\mathrm{X}, \mathrm{Y}) = \frac{\mathrm{HSIC}(x, y)}{\sqrt{\mathrm{HSIC}(x, x)\mathrm{HSIC}(y, y)}}. \quad (2)$$

Here, HSIC determines whether two distributions are independent. Formally, let $K_{ij} = k\left(\mathrm{x}_i, \mathrm{x}_j\right)$ and $L_{ij} = l\left(\mathrm{y}_i, \mathrm{y}_j\right)$, where $k$ and $l$ are two kernels. HSIC is defined as

$$\mathrm{HSIC}(K, L) = \frac{1}{(n-1)^2}\mathrm{Tr}(KHLH), \quad (3)$$

where $H$ denotes the centering matrix (*i.e.*, $H_n = I_n - \frac{1}{n}\mathbf{1}\mathbf{1}^T$). For kernel selections of $k$ and $l$, we find that different kernels (RBF, polynomial, and linear) give similar results and rankings, and the RBF kernel contributes to the distinguishability of quantitative results. Therefore, RBF kernel is used for all experiments, and the bandwidth is set as a fraction of the median distance between examples [32]. These metrics are compared in a consistent setting for fair comparison, more implementation details and comparisons on CKA are given in *Supplementary Material*.

## 2.3 Benchmarks and Analysis Approaches

**Benchmarks.** In order to analyze various factors of measurement, we also collect multiple generators to produce synthesized images. To be more specific, we employ state-of-the-art generative models trained on various datasets for comparison in Tab. 1. We download corresponding publicly available models for comparison. Unless otherwise specified, all of these models are compared in a consistent setting.

**Visualization tool.** To qualitatively compare where these feature extractors "focus on", we employ the visualization technique of [35] to localize the regions that contribute the most to the similarity index. The highlighted regions reveal the most relevant parts of an image regarding the measurement results. Accordingly, larger highlighted regions indicate that more visual semantics are involved

Table 1: **Generative models used in our study**. Publicly available models are gathered for evaluation.

| Method | Year | Training Datasets |
|--------|------|-------------------|
| StyleGAN2 [29] | 2020 | FFHQ, LSUN Church |
| BigGAN, BigGAN-deep [5] | 2019 | ImageNet |
| ADM, ADM-G [14] | 2021 | ImageNet |
| Projected-GAN [50] | 2021 | FFHQ, LSUN Church |
| InsGen [66] | 2021 | FFHQ |
| StyleGAN-XL [66] | 2022 | FFHQ, ImageNet |
| Aided-GAN [33] | 2022 | LSUN Church |
| EqGAN [62] | 2022 | FFHQ, LSUN Church |
| Diffusion-GAN [64] | 2022 | LSUN Church |
| DiT [45], BigRoC [18] | 2022 | ImageNet |
| GigaGAN [27], DG-Diffusion [31], MDT [19] | 2023 | ImageNet |

in the evaluation. Note that generating a realistic image requires all parts, even each pixel, to be well-synthesized. Thus, a metric that focuses on more visual regions appears to be more dependable.

**Histogram matching attack.** We employ the histogram matching [35] to attack the system to investigate the robustness of the measurement results. Concretely, a subset is selected from a superset by matching the class distribution of the synthesized set with that of the real set. As pointed out by [35], the synthesis performance could be substantially improved using the chosen subset. We thus prepare two distinct sets of synthesized images. One reference set ("Random") is produced by generating images randomly, and the other set ("Chosen$_I$") is carefully curated by matching the class distribution histogram of the supervised Inception-V3 [56]. The matched histograms are available in *Supplementary Material*. Since the generator and real data remain unaltered, the evaluation should keep consistent, and any performance gains directly indicate the unreliability of a given extractor.

**Human judgment.** In order to examine the correlation between the evaluation system and human perceptual judgment, we conduct extensive user studies employing two strategies. Firstly, to benchmark the synthesis quality of various generative models, we prepare a substantial number of randomly generated images (*i.e.,* $5K$), and ask 100 individuals to assess the photorealism of these images. The final scores are averaged across all 100 participants. Secondly, to qualitatively compare two paired generative models with similar quantitative performances (*e.g.,* Projected-GAN [50] and Aided-GAN [33] on LSUN Church dataset in Sec. 4.1), we prepare groups of paired images generated by different models and ask 100 individuals to assess the perceptual quality of paired images. More details of our user studies can be found in *Supplementary Material*. In this way, we could obtain reliable and consistent human judgments, facilitating better investigation with the evaluation system.

## 3 Analysis on Representation Spaces and Distributional Distances

In this section, we investigate the potential impacts of the representation space and distributional distances with respect to the final similarity index. Concretely, Sec. 3.1 presents the study of extractors that define the representation spaces, followed by the analysis of distributional distances in Sec. 3.2.

### 3.1 Representation Spaces

Prior works [35, 36, 41, 26] have demonstrated that the most commonly used feature extractor *i.e.,* Inception-V3 [56], could hardly reflect the exact improvement of synthesis due to its limited consideration of semantics. We thus conduct a comprehensive study by incorporating various models that differ in *supervision signals* and *architectures*, to identify which or which set of extractors serve as reliable feature extractors for synthesis comparison.

**Distinct feature extractors with different architectures yield varying semantic areas of focus.** Fig. 1 shows the highlighted regions that contribute most significantly to the measurement results. Obviously, CNN-based extractors consistently emphasize concentrated regions with or without manual labels for pre-training, including limited semantics. Specifically, CNN-based extractors remain to highlight objects (*e.g.,* microphone, hat, and sunglasses), rather than the main focus of the evaluation domains (*i.e.,* Human Faces here). In contrast, ViT-based extractors capture larger regions that encompass more synthesis details and semantics. This observation aligns with the finding that ViTs possess a global and expansive receptive field compared to the local receptive field of

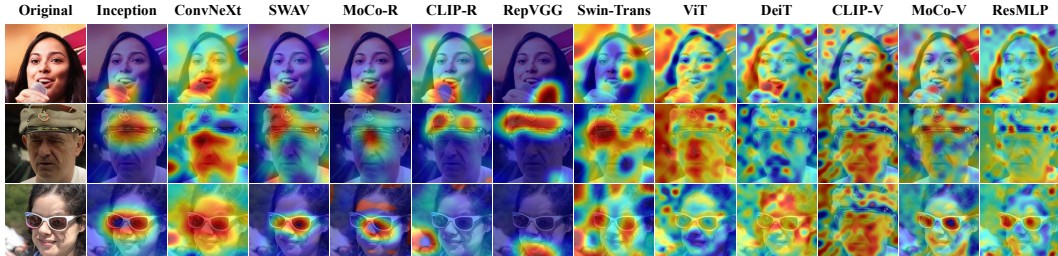

| Original | Inception | ConvNeXt | SWAV | MoCo-R | CLIP-R | RepVGG | Swin-Trans | ViT | DeiT | CLIP-V | MoCo-V | ResMLP |
|---|---|---|---|---|---|---|---|---|---|---|---|---|

Figure 1: **Heatmaps from extractors with various architectures.** CNN-based extractors (*i.e.,* Inception [56], ConvNeXt [40], SWAV [6], MoCo-R [9], CLIP-R [46], and RepVGG [15]) focus on objects whereas ViT-based (*i.e.,* Swin-Transformer [39], ViT [16], DeiT [60], CLIP-V [46], and MoCo-V [9]) and MLP-based (*i.e.,* ResMLP [61]) ones pour attention on wider areas.

Table 2: **Quantitative comparison results of Fréchet Distance (FD$_{\downarrow}$) on FFHQ dataset**. "Random, Chosen$_I$" respectively represent the synthesized distribution of randomly generated and matching the class prediction of Inception-V3. Moreover, "-R" and "-V" respectively denote the architecture of ResNet and ViT. ($_{\downarrow}$) indicates the results are hacked by the histogram matching mechanism. Notably, the values across different rows are not comparable and we report the stds of three testings to better illustrate the numerical fluctuation of various extractors towards the histogram attack. The results of Fréchet Distance (FD) on ImageNet can be found in *Supplementary Material*.

| Model | Inception | ConvNeXt | SWAV | MoCo-R | RepVGG | CLIP-R |
|---|---|---|---|---|---|---|
| Random | $2.81_{\pm 0.01}$ | $78.03_{\pm 0.10}$ | $0.13_{\pm 0.002}$ | $0.24_{\pm 0.003}$ | $129.61_{\pm 0.41}$ | $10.34_{\pm 0.06}$ |
| Chosen$_I$ | $2.65_{\pm 0.01\downarrow}$ | $78.19_{\pm 0.11}$ | $0.13_{\pm 0.002}$ | $0.24_{\pm 0.003}$ | $129.67_{\pm 0.39}$ | $10.36_{\pm 0.08}$ |

| Model | Swin | ViT | DeiT | CLIP-V | MoCo-V | ResMLP |
|---|---|---|---|---|---|---|
| Random | $142.87_{\pm 0.12}$ | $15.11_{\pm 0.09}$ | $437.80_{\pm 0.14}$ | $1.06_{\pm 0.01}$ | $7.32_{\pm 0.03}$ | $99.11_{\pm 0.06}$ |
| Chosen$_I$ | $140.01_{\pm 0.12\downarrow}$ | $15.11_{\pm 0.10}$ | $430.81_{\pm 0.16\downarrow}$ | $1.06_{\pm 0.01}$ | $7.40_{\pm 0.03}$ | $95.36_{\pm 0.06\downarrow}$ |

CNN-based extractors [47, 65, 16]. Furthermore, the ViT-based and CNN-based extractors appear to complement each other, with the former capturing broader regions and the latter focusing on specific objects with high density.

**Multiple extractors incorporate more visual semantics in a complementary manner.** When reproducing the whole data distribution, generative models are required to synthesize not only the main objects but also the background, texture, and intricate details. Similarly, the extractors should strive to capture more regions of given images to approximate the data distribution, enabling better visual perception. However, the above observation regarding the heatmaps of different extractors reveals that each individual extractor could only capture partial semantics of the entire image for measurement, inadequately reflecting the overall synthesis performance. Consequently, various extractors with different architectures should be considered since they could involve more semantics, enhancing the reliability of the evaluation.

**Extractors that are vulnerable to the histogram matching attack are not reliable for evaluation.** Prior study [35] has highlighted that extractors focusing on limited semantics may be susceptible to the histogram matching attack, undermining the trustworthiness of the evaluation. Motivated by this, we investigate the robustness of the above extractors toward the attack as they attach different importance to the visual concepts. Concretely, we obtain the publicly available generator of StyleGAN[2] [30] and calculate the Fréchet Distance (FD) results on the FFHQ dataset. We compare two evaluated sets: one is randomly generated using the model, while the other is chosen by matching the predicted class distribution of Inception-V3 (the matched histograms are provided in *Supplementary Material*).

Tab. 2 shows the quantitative results. Comparing the performances of the random set and chosen set, we could tell that certain extractors, regardless of their architectures (*e.g.,* CNN-based Inception-V3 [56], ViT-based Swin-Transformer [39], and MLP-based ResMLP [61]), are susceptible to the histogram matching attack. For instance, the FD score of Inception shows an improvement of 5.7% when the chosen set is used, and ResMLP exhibits a 3.8% improvement (More quantitative

---

[2]https://github.com/NVlabs/stylegan3

and qualitative results of MLP-based extractors gMLP [37] and MLP-mixer [59] are shown in *Supplementary Material*). Namely, the FD score could be improved without making any changes to the generators, aligning with the observations of [35]. We thus filter out the extractors that are vulnerable to the attack since their rankings can be manipulated without actual improvement to the generative models.

**Extractors that define similar representation spaces are redundant.** So far, we have demonstrated the importance of considering multiple extractors for a more comprehensive evaluation and filtered out the extractors that are susceptible to the histogram attack. However, it is also crucial to avoid redundancy among the remaining extractors, as they may define similar representation spaces. To address this, we examine the correlation between representation spaces across different feature extractors, following the approach outlined in [32]. Specifically, a significant number of images ($10K$ images from ImageNet) are fed into these extractors to compute their correspondence. After calculating the similarity matrix, we can further filter out extractors that define homogeneous representation spaces (the similarity analysis is presented in *Supplementary Material*). The remaining extractors are presented in the table below. These extractors 1) capture rich semantics in a

| | |
|---|---|
| **CNN-based** | ConvNeXt [40], SWAV [6], RepVGG [15] |
| **ViT-based** | CLIP-ViT [46], MoCo-ViT [9], ViT [16] |

complementary way, 2) are robust toward the histogram matching attack, and 3) define meaningful and distinctive representation spaces. Besides, both CNN-based and ViT-based extractors are considered and all of them have demonstrated strong performance for the pre-defined and downstream tasks, facilitating more comprehensive and reliable evaluation. Notably, the selection of self-supervised extractors SWAV, CLIP-ViT, and MoCo-ViT agrees with previous studies [41, 35, 3].

**The selected extractors can serve as reliable tools for synthesis evaluation.** In order to investigate the reliability of the selected extractors, we employ them to test the synthesis quality of representative generative models on the ImageNet dataset. Tab. 3 presents the quantitative FD scores of various extractors. Although their results differ in numerical scales, they consistently indicate that StyleGAN-XL outperforms both BigGAN and BigGAN-deep by a significant margin, and BigGAN-deep performs slightly better than BigGAN. Such observation also agrees with the human judgment in Tab. 5. Overall, the consistent trends observed across different extractors and the alignment with human judgment confirm that the selected extractors are reliable for comparing the synthesis quality.

### 3.2 Distributional Distances

After the study of feature extractors, we shift our focus to another essential component of measurement, *i.e.,* the distributional distance $d(\cdot)$. Besides Fréchet Distance (FD), we also incorporate Centered Kernel Alignment (CKA) for our investigation.

**CKA provides normalized distances *w.r.t* numerical scales in variable representation spaces.** Tab. 5 demonstrates the quantitative results of Centered Kernel Alignment (CKA). Unlike the Fréchet Distance (FD) scores that exhibit significant fluctuations across various extractors, the CKA scores demonstrate remarkable stability when evaluated in different representation spaces. For instance, the FD score of BigGAN on MoCo-ViT is 238.78 whereas 3.35 on CLIP-ViT, making it challenging to combine the results of different extractors. By contrast, the stability of CKA scores allows the ability to combine results from multiple extractors (*i.e.,* average) for better comparison.

**CKA demonstrates great potential for quantitative comparison and combination across hierarchical layers.** Here we investigate the distributional distances across various layers of the neural network, as these layers typically extract multi-level features that span from high-level semantics to low-level details. Accordingly, considering hierarchical features can provide a more comprehensive measurement. The left part of Tab. 4 presents the qualitative results of different layers' heatmaps. We can observe that different layers indeed extract different semantics, highlighting the importance of considering hierarchical features in evaluation. Additionally, we provide the quantitative results of FD and CKA in the right part of Tab. 4. Still, the FD scores of various layers fluctuate dramatically, *e.g.,* the FD results of $0.60$ in $\text{Layer}_1$ while $104.10$ in $\text{Layer}_4$. Differently, the CKA results from hierarchical layers are comparable and the overall score could be derived by averaging multi-level scores. Importantly, the overall score still reflects synthesis quality consistently and reliably.

**CKA shows satisfactory sample-efficiency and stability under different number of samples.** Typically, the synthesis quality are measured between real and synthesized distributions, where the

Table 3: **Quantitative comparison results of Fréchet Distance (FD$_\downarrow$) on ImageNet dataset**. [†] scores are quoted from the original paper and other results are tested three times. Notably, the values across different columns are not comparable.

| Model | FID[†] | ConvNeXt | RepVGG | SWAV | ViT | MoCo-ViT | CLIP-ViT | Overall | User study |
|---|---|---|---|---|---|---|---|---|---|
| BigGAN [5] | 8.70 | 140.04 | 67.53 | 1.12 | 29.95 | 238.78 | 3.35 | N/A | 53% |
| BigGAN-deep [5] | 6.02 | 102.26 | 58.85 | 0.87 | 23.98 | 85.83 | 3.22 | N/A | 55% |
| StyleGAN-XL [51] | 1.81 | 19.22 | 15.93 | 0.18 | 8.51 | 29.38 | 1.85 | N/A | 67% |

Table 4: **Heatmaps from various semantic levels on FFHQ dataset (*left*) and Fréchet Distance (FD$_\downarrow$) and Centered Kernel Alignment (CKA$_\uparrow$) scores on ImageNet dataset (*right*)**. CLIP-ViT serves as the feature extractor here, more results can be found in *Supplementary Material*.

| Model | BigGAN | | BigGAN-deep | | StyleGAN-XL | |
|---|---|---|---|---|---|---|
| Layer | FD$_\downarrow$ | CKA$_\uparrow$ | FD$_\downarrow$ | CKA$_\uparrow$ | FD$_\downarrow$ | CKA$_\uparrow$ |
| Layer$_1$ | 0.60 | 99.06 | 0.54 | 98.95 | 0.05 | 99.84 |
| Layer$_2$ | 7.45 | 86.89 | 5.58 | 90.09 | 0.77 | 91.06 |
| Layer$_3$ | 30.24 | 82.80 | 23.55 | 83.63 | 6.11 | 85.75 |
| Layer$_4$ | 104.10 | 80.13 | 81.02 | 81.05 | 35.77 | 83.55 |
| Overall | N/A | 87.22 | N/A | 88.43 | N/A | 90.05 |

whole training data is used as the real distribution and $50K$ generated images as the synthesized, regardless of how many samples contained in the training data. However, when evaluating on the large-scale datasets (*e.g.,* $1.28$ million images for ImageNet), $50K$ images may be insufficient to represent the entire distribution. Therefore, we study the impacts of the amount of generated samples. Concretely, we prepare several synthesized sets with different numbers of samples and calculate their distances to the real distribution (more details are presented in *Supplementary Material*).

Fig. 2 demonstrates the curves of FD and CKA scores evaluated under different data regimes on the FFHQ dataset. Obviously, the FD scores can be drastically improved by synthesizing more data regardless of different extractors until sufficient samples ($\sim 100K$) are used, whereas the CKA scores are stable under different data regimes. Moreover, CKA could measure the distributional distances precisely with only $5K$ synthesized samples, suggesting significant sample efficiency. Such observations demonstrate CKA's impressive adaptability toward the amount of synthesized data.

**Developing a reliable and comprehensive measurement system for synthesis evaluation.** Overall, a set of feature extractors that 1) are robust to the histogram matching attack, 2) capture sufficient semantics, and 3) define distinctive representation spaces could serve as reliable extractors for synthesis comparison. Together with a bounded distance (*i.e.,* CKA) that is comparable across various representation spaces and hierarchical layers, as well as enjoys satisfactory sample efficiency. These two essential components constitute a reliable system to deliver the distributional discrepancy.

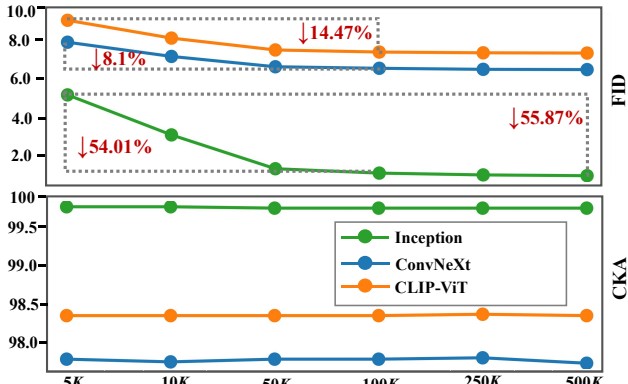

Figure 2: **Fréchet Distance (FD) and Centered Kernel Alignment (CKA) scores evaluated under various data regimes on FFHQ dataset.** FID scores are scaled for better visualization. $\downarrow$ denotes the results fluctuate downward. The percentages represent the magnitude of the numerical variation. The curve of KID [4], Precision and Recall [49] can be found in *Supplementary Material*.

Table 5: **Quantitative comparison results of Centered Kernel Alignment (CKA$_\uparrow$) on ImageNet dataset**. [†] scores are quoted from the original paper and others are tested three times. To make our results more trivial to parse, we visualize the correlation between different metrics the human evaluation results in the *Supplementary Material*.

| Model | FID[†] | ConvNeXt | RepVGG | SWAV | ViT | MoCo-ViT | CLIP-ViT | Overall | User study |
|---|---|---|---|---|---|---|---|---|---|
| ICGAN [7] | 15.60 | 62.65 | 72.25 | 86.10 | 97.04 | 77.99 | 92.16 | 81.37 | 32% |
| ADM [14] | 10.94 | 63.12 | 72.90 | 87.71 | 97.39 | 78.69 | 92.92 | 82.12 | 45% |
| BigGAN [5] | 8.70 | 63.89 | 74.62 | 87.94 | 97.60 | 79.60 | 93.25 | 82.82 | 53% |
| C-ICGAN [7] | 7.50 | 62.53 | 72.20 | 86.12 | 97.05 | 78.01 | 92.08 | 81.33 | 31% |
| BigGAN-deep [5] | 6.95 | 64.97 | 76.45 | 88.31 | 97.77 | 80.27 | 94.11 | 83.65 | 55% |
| Guided-ADM [14] | 4.59 | 65.99 | 78.99 | 89.44 | 98.13 | 80.46 | 94.96 | 84.66 | 57% |
| BigRoc [18] | 3.69 | 67.86 | 79.48 | 89.93 | 98.23 | 82.25 | 96.07 | 85.64 | 65% |
| GigaGAN [27] | 3.45 | 68.01 | 79.93 | 90.15 | 98.34 | 82.40 | 96.52 | 85.89 | 65% |
| DG-Diffusion [31] | 3.18 | 68.22 | 80.06 | 90.56 | 98.46 | 82.51 | 96.88 | 86.12 | 66% |
| StyleGAN-XL [51] | 2.30 | 68.75 | 80.28 | 91.54 | 98.52 | 82.64 | 97.41 | 86.52 | 67% |
| DiT [45] | 2.27 | 68.94 | 80.65 | 91.03 | 99.05 | 82.90 | 97.08 | 86.61 | 67% |
| MDT [19] | 1.79 | 69.64 | 81.68 | 91.78 | 99.43 | 83.43 | 98.19 | 87.36 | 69% |

Table 6: **Quantitative comparison results of Centered Kernel Alignment (CKA$_\uparrow$) on FFHQ dataset**. [†] scores are quoted from the original paper and others are tested three times.

| Model | FID[†] | ConvNeXt | RepVGG | SWAV | ViT | MoCo-ViT | CLIP-ViT | Overall | User study |
|---|---|---|---|---|---|---|---|---|---|
| StyleGAN2 [29] | 3.66 | 92.64 | 62.10 | 99.55 | 99.32 | 98.59 | 97.46 | 91.61 | 45% |
| Projected-GAN [50] | 3.39 | 92.34 | 61.62 | 99.37 | 98.99 | 98.81 | 97.31 | 91.41 | 39% |
| InsGen [66] | 3.31 | 94.17 | 65.72 | 99.60 | 99.36 | 98.90 | 97.75 | 92.58 | 58% |
| EqGAN [62] | 2.89 | 94.54 | 64.36 | 99.69 | 99.49 | 99.10 | 98.61 | 92.63 | 62% |
| StyleGAN-XL [51] | 2.19 | 93.78 | 67.59 | 99.68 | 99.49 | 99.25 | 97.33 | 92.85 | 66% |

## 4 Benchmark Existing Generative Models

Based on the findings regarding the feature extractors and distributional distances, we construct a new synthesis valuation system. Concretely, our system leverages a set of models with both CNN and ViT architectures as feature extractors, namely, CNN-based ConvNeXt [40], RepVGG [15], SWAV [6] and ViT-based ViT [16], MoCo-ViT [9], CLIP-ViT [46], with which more comprehensive evaluation could be accomplished. Further, Centered Kernel Alignment (CKA) serves as the similarity indicator.

Accordingly, in this part we re-evaluate and compare the progress of existing generative models with our measurement system. Concretely, the latest generative models are re-evaluated with our system in Sec. 4.1, followed by our discussion about the diffusion models and GANs in Sec. 4.2. Notably, user studies are conducted for investigating the correlation between our system and human judgment.

### 4.1 Comparison on Existing Generative models

In order to investigate the actual improvement of existing generative models, we collect multiple publicly available generators trained on several popular benchmarks (*i.e.,* FFHQ, LSUN Church, and ImageNet) for comparison. Benefiting from the impressive sample efficiency of CKA, we generate $50K$ images as the synthesized distribution and use the whole datasets as the real distribution. Results from various selected extractors and their averaged scores are reported for a thorough comparison.

**Our system can measure the synthesis quality in a consistent and reliable way.** Tab. 5, Tab. 7, and Tab. 6 respectively demonstrate the quantitative results of different generative models on the ImageNet, LSUN Church, and FFHQ datasets. In most cases, CKA scores from various extractors and the overall scores provide a consistent ranking with FID scores, as well as agree with human perceptual judgment. These results suggest that our new metric could precisely measure the synthesis quality. However, for the Projected-GAN [50] and StyleGAN2 [29] (*resp.,* Aided-GAN [33]) evaluated on FFHQ (*resp.,* LSUN Church) dataset in Tab. 6 (*resp.,* Tab. 7), our evaluation system gives the opposite ranking to the FID. Namely, the quantitative results of StyleGAN2 (*resp.,* Aided-GAN) are determined better than that of Projected-GAN under our evaluation, whereas the FID scores vote Projected-GAN for the better one. Additionally, the performances of ICGAN [7] and class-conditional ICGAN on ImageNet in Tab. 5 are identified basically the same by our metric, while FID scores indicate that the class-conditional one significantly surpasses the unconditional one.

Table 7: **Quantitative comparison results of Centered Kernel Alignment (CKA$_\uparrow$) on LSUN Church dataset**. [†] scores are quoted from the original paper and others are tested three times.

| Model | FID[†] | ConvNeXt | RepVGG | SWAV | ViT | MoCo-ViT | CLIP-ViT | Overall | User study |
|---|---|---|---|---|---|---|---|---|---|
| StyleGAN2 [29] | 3.86 | 96.99 | 67.43 | 98.70 | 97.68 | 89.18 | 76.25 | 87.71 | 69% |
| Diffusion-GAN [64] | 3.17 | 97.15 | 69.28 | 99.22 | 97.70 | 90.25 | 79.06 | 88.78 | 71% |
| EqGAN [66] | 3.02 | 97.34 | 71.12 | 99.36 | 98.09 | 90.64 | 80.26 | 89.47 | 73% |
| Aided-GAN [33] | 1.72 | 97.91 | 75.63 | 99.42 | 99.62 | 92.56 | 81.69 | 91.14 | 77% |
| Projected-GAN [50] | 1.59 | 96.89 | 72.91 | 97.98 | 98.09 | 91.43 | 78.63 | 89.34 | 72% |

Table 8: **Fine-grained investigation of human judgment**. Percentages indicate the ratio of generated images that are considered to be more plausible.

| Dataset | FFHQ | | LSUN Church | | ImageNet | |
|---|---|---|---|---|---|---|
| Model | Projected-GAN | StyleGAN2 | Projected-GAN | Aided-GAN | ICGAN | Conditional-ICGAN |
| User Preference | 45% | 55% | 43% | 57% | 50% | 50% |

In order to compare the performance of these models in a fine-grained way, we further perform paired-wise human evaluation. Specifically, groups of paired images synthesized by different models (*e.g.,* StyleGAN2 and Projected-GAN on FFHQ dataset) are randomly picked for visual comparison. Then, presented with two sets of images produced by different models, users are asked to determine which set of images is more plausible.

**Our measurement system provides the right rankings and better correlations with human visual judgment.** Tab. 8 presents the quantitative results of human visual comparison. Observably, the synthesis quality of StyleGAN (*resp.* Aided-GAN) is more preferred by human visual judgment. That is, our measurement system produces the same rankings as the human perceptual evaluation, demonstrating the reliability of our metric. Moreover, our metric's indication that there's no significant gap between ICGAN and class-conditional ICGAN is also verified by the human evaluation. Considering the perceptual null space of Inception-V3, one possible reason for the FID performance gains of Projected-GAN might be the usage of pre-trained models, which is also identified by [35]. By contrast, our measurement produces the right rankings and agrees well with human evaluation, reflecting the actual synthesis quality in a comprehensive and reliable way.

### 4.2 Comparison between GANs and Diffusion Models

Diffusion models [23, 54, 55, 14, 45, 48, 19, 70, 42, 24] have demonstrated significant advancements in visual synthesis and became the new trend of generative models recently. Benefiting from the reliability of our new measurement system, here we perform a comprehensive comparison between GANs and diffusion models. Specifically, we report the FID and the overall CKA scores, as well as human judgment for quantitative comparison. Additionally, the model parameters and the synthesis speed (tested on an A100 GPU) are also included.

Tab. 9 presents the quantitative results. Obviously, diffusion model (*i.e.*, DiT) obtains comparable results with GAN (*i.e.*, StyleGAN-XL), yet with much more parameters (*i.e.,* $675M$ *vs.* $166.3M$). Moreover, diffusion models usually require extra inference time to obtain realistic images. Such comparisons reveal that GANs achieve better trade-offs between efficiency and synthesis quality, and designing computation-efficient diffusion models is essential for the community.

### 4.3 Comparing the performance of image-to-image translation

In order to testify the compatibility of our metric, here we employ our measurement system to evaluate the performance of image-to-image translation. We collect publicly available image-to-image translation models that are officially released to translate images from one domain to another domain for evaluation. Specifically, three translation benchmarks are involved here, namely Horse-to-Zebra [57, 71, 43], Cat-to-Dog [68, 43], and Dog-to-Cat [68, 25]. For each benchmark, we translate the tested images to the target domain following the original experimental settings. Then we compute the distributional discrepancies between the translated images and the real target images. Tab. 10 presents the quantitative results of the evaluated three image-to-image translation benchmarks. It can

Table 9: **Quantitative comparison results of diffusion models and GANs on ImageNet dataset**. [†] scores are quoted from the original paper and [‡] results are overall scores measured by our system.

| Model | FID[†] | CKA[‡] | User | #Params | Sec/$K$img (s) |
|---|---|---|---|---|---|
| BigGAN | 8.70 | 82.82 | 53% | 158.3 $M$ | 33.6 |
| BigGAN-deep | 6.95 | 83.65 | 55% | 85 $M$ | 27.6 |
| StyleGAN-XL | 2.30 | 86.52 | 67% | 166.3 $M$ | 64.8 |
| ADM | 10.94 | 82.12 | 45% | 500 $M$ | 17274 |
| Guided-ADM | 4.59 | 84.66 | 57% | 554 $M$ | 17671 |
| DiT | 2.27 | 86.61 | 67% | 675 $M$ | 3736.8 |

be seen from these results that CKA provides consistent ranks with FID among various extractors, and the averaged score can reflect the performance of different image translation models. For instance, the performance of CUT [43] on Horse-to-Zebra is identified better than that of CycleGAN [71] by both FID and our proposed metric. And the qualitative results in the original paper of CUT [43] also suggest that the performance of CUT surpasses CycleGAN. That is, our measurement system can provide a reliable evaluation under such settings. This indicates that our measurement system can also be used for evaluating the performance of image translation tasks.

Table 10: **Quantitative comparison results of Centered Kernel Alignment (CKA$_\uparrow$) on Image-to-Image translation tasks**.

| **Horse-to-Zebra dataset** | | | | | | | |
|---|---|---|---|---|---|---|---|
| Model | FID | ConvNeXt | RepVGG | SWAV | ViT | MoCo-ViT | CLIP-ViT | Overall |
| CycleGAN [71] | 83.32 | 73.55 | 88.67 | 85.82 | 83.96 | 74.72 | 73.74 | 80.08 |
| AttentionGAN [57] | 76.05 | 75.59 | 91.73 | 86.37 | 85.16 | 76.65 | 75.49 | 81.83 |
| CUT [43] | 51.29 | 78.48 | 93.22 | 88.83 | 87.84 | 78.75 | 77.36 | 84.08 |

| **Cat-to-Dog** | | | | | | | |
|---|---|---|---|---|---|---|---|
| Model | FID | ConvNeXt | RepVGG | SWAV | ViT | MoCo-ViT | CLIP-ViT | Overall |
| CUT [43] | 74.95 | 84.93 | 78.75 | 88.83 | 84.31 | 93.56 | 70.91 | 83.55 |
| GP-UNIT [68] | 60.96 | 90.45 | 87.79 | 94.05 | 90.12 | 95.91 | 75.32 | 88.94 |

| **Cat-to-Dog** | | | | | | | |
|---|---|---|---|---|---|---|---|
| Model | FID | ConvNeXt | RepVGG | SWAV | ViT | MoCo-ViT | CLIP-ViT | Overall |
| GP-UNIT [68] | 31.66 | 79.58 | 78.18 | 96.79 | 86.93 | 93.92 | 77.42 | 85.47 |
| MUNIT [25] | 18.88 | 84.87 | 84.11 | 98.51 | 88.11 | 95.95 | 86.10 | 89.61 |

## 5  Conclusion

This work revisits the evaluation of generative models from the perspectives of the feature extractor and the distributional distance. Through extensive investigation regarding the potential contribution of various feature extractors and distributional distances, we identify the impacts of several potential factors that contribute to the final similarity index. With these findings, we construct a new measurement system that provides a more comprehensive and holistic comparison for synthesis evaluation. Importantly, our system could present more consistent measurements with human judgment, enabling more reliable evaluation.

**Discussion.** Despite a comprehensive investigation, our study could still be extended in several aspects. For instance, the impacts of different low-level image processing techniques (*e.g.,* resizing) could be identified since they also play an important role in synthesis evaluation [44]. Besides, comparing datasets with various resolutions could be further studied. Nonetheless, our study could be considered an empirical revisiting towards the paradigm of evaluating generative models. We hope this work could inspire more fascinating works of synthesis evaluation and provide potential insight to develop more comprehensive evaluation protocols. We will also conduct more investigation on the unexplored factors and compare more generative models with our system.

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

# A  Appendix

This *Supplementary Material* is organized as follows: appendix B provides the implementation details of our experiments, appendix C demonstrates how human visual judgment is performed and presents the interface of our user study. appendix D presents additional quantitative and qualitative results, including 1) comparison results with more metrics (KID, Precision and Recall), 2) more results of MLP-based extractors, 3) the similarities between various representation spaces, as well as 4) more hierarchical results from different semantic levels of various extractors. Finally, in appendix E, we provide 2D plots of the correlation between different metrics with human visual judgment to make our results easier to parse.

# B  Implementation Details

## B.1  Datasets

**FFHQ** [58] contains unique $70,000$ human-face images with large variations in terms of age, ethnicity, and facial expressions. We employ the resolution of $256 \times 256 \times 3$ for our experiments.

**ImageNet** [13] includes $1,280,000$ images with $1,000$ classes of different objects such as goldfish, bow tie, etc. All experiments on ImageNet are performed with the resolution of $256 \times 256 \times 3$ unless otherwise specified.

**LSUN Church** [69] consists of $126,227$ images of the church, varies in the background, perspectives, etc. We employ the resolution of $256 \times 256 \times 3$ for our experiments.

## B.2  Experimental Settings and Hyperparameters

**Kernel selection**. We consistently employ the RBF kernel

$$K(\mathbf{x_i}, \mathbf{x_j}) = \exp(-\frac{\|\mathbf{x_i} - \mathbf{x_j}\|^2}{2\sigma^2})$$

for calculating the CKA. The bandwidth $\sigma$ is set as a fraction of the median distance between examples. In practice, three commonly used kernels could be employed for calculation, namely linear, polynomial, and RBF kernels. In order to investigate their difference, three publicly available models with clear performance margins are collected for evaluation. Concretely, we gather models of InsGen [66] trained on FFHQ with different data regimes (*i.e.,* 2K, 10K, 140K), the ranking of their synthesis quality is clear and reasonable.

Tab. A1 demonstrates the quantitative results of CKA with different kernels. Obviously, these kernels give similar results and rankings. However, the RBF kernel contributes to the distinguishability of quantitative results, making the results more comparable. Consequently, the RBF kernel is employed in our experiments.

Table A1: **CKA results with different kernels.** The publicly available models are gathered for comparison. $^\dagger$ results are quoted from the original paper.

| Kernel | InsGen-2k | InsGen-10k | InsGen-140k |
|---|---|---|---|
| FID$^\dagger$ | 11.92 | 4.90 | 3.31 |
| Linear | 99.83 | 99.93 | 99.98 |
| Poly | 99.58 | 99.87 | 99.92 |
| RBF | 95.72 | 98.65 | 99.10 |

**ViT features calculation**. The feature maps of ViT-based extractors are three-dimensional tensors (N, W, C), where W contains the global token and local features. The global token captures the same semantic information as the local features. Thus the global taken features are used for computation in implementation. Tab. A2 shows the comparison results of using local features and the global token. Consistently, they give similar results and rankings, so we use the global token for calculation in our experiments.

**Feature normalization**. In practice, the activations of features play an essential role in computing the similarity index. Namely, the quantitative results would be dominated by a few activations with large

Table A2: **CKA results with different features for calculation.**

| Metrics | InsGen-2k | InsGen-10k | InsGen-140k |
|---|---|---|---|
| Local Features | 96.62 | 97.42 | 97.38 |
| Global Token | 97.46 | 97.88 | 97.93 |

Table A3: **CKA scores with different normalization techniques**.

| Metrics | InsGen-2k | InsGen-10k | InsGen-140k |
|---|---|---|---|
| $CKA_{No}$ | 97.46 | 97.88 | 97.93 |
| $CKA_{L1}$ | 96.62 | 98.91 | 99.33 |
| $CKA_{L2}$ | 96.62 | 98.91 | 99.32 |
| $CKA_{Softmax}$ | 95.72 | 98.65 | 99.10 |

peaks, neglecting other correlation patterns [63]. To investigate the activations of our self-supervised extractors, we visualize the activations of different samples and their statistics.

Fig. A1 and Fig. A2 respectively illustrate the activation of different samples and their statistics. Obviously, there are several peaks in the activations. And these peaks may dominate the similarity index as they are substantially larger than other activations. To mitigate the peaks and create a more uniform distribution, we employ the softmax transformation [63] to normalize the features. Such operation smooths the activations while maintaining the original distributional information of features. Thus the similarity index remains consistent to deliver the distribution discrepancy. Besides the softmax transformation, we also compare the behavior of different normalization techniques (*i.e.,* $L_1$ and $L_2$ normalization).

Tab. A3 demonstrate the quantitative results with different normalization techniques. They consistently provide similar results and rankings, and the softmax transformation ameliorates the peaks more significantly, providing more comparable results. Consequently, we adopt Softmax normalization in our experiments.

**Histogram matching.** In order to investigate the robustness of the measurement system, we employ the histogram matching [35] to attack the system. To be specific, a subset with a considerable number (*e.g.,* $50K$) of images is chosen as the referenced distribution, and the corresponding class distribution is predicted by a given classifier (*i.e.,* Inception-V3 [56]). With the guidance of the classifier, the generator is encouraged to produce a synthesis distribution that matches the predicted class distribution of real images. Recall that the generator used to produce these synthesized distributions stays unchanged, thus a robust measurement system should give consistent similarities between the randomly generated and the matched distribution.

Fig. A3 provides the class distribution of real and synthesized FFHQ images predicted by Inception-V3. Obviously, the class distribution of the matched distribution is well-aligned with the predicted real distribution.

**Sample-efficiency.** In order to investigate the impacts of the number of synthesized samples, we compute the distributional distances between the real distribution with synthesized distributions with various numbers of generated images. Concretely, FFHQ (with $70K$ images) and ImageNet (with 1.28 million images) are investigated for universal conclusions. For both datasets, we synthesis $500K$ images as candidate, and randomly choose $5K$, $10K$, $50K$, $100K$, $250K$, and $500K$ images as the synthesized distribution for computing the metrics. The entire training data is utilized as the real distribution, and the publicly accessible models on FFHQ[3] and ImageNet[4] are employed.

**The curve of FD and CKA under various data regimes on ImageNet dataset** is shown in Fig. A4. Consistent with the aforementioned results in the main paper, CKA could measure the distributional distances precisely with only $5K$ samples, whereas FID fails to deliver the actual measurement until sufficient samples are used. That is, CKA could give reliable results even when limited data is given, suggesting impressive sample efficiency. Equipped with the bounded quantitative results and consistency under different data regimes, as well as the robustness to the histogram matching attack, CKA outperforms FID as a reliable distance for delivering the distributional discrepancy.

---

[3]https://github.com/NVlabs/stylegan3
[4]https://github.com/autonomousvision/stylegan-xl

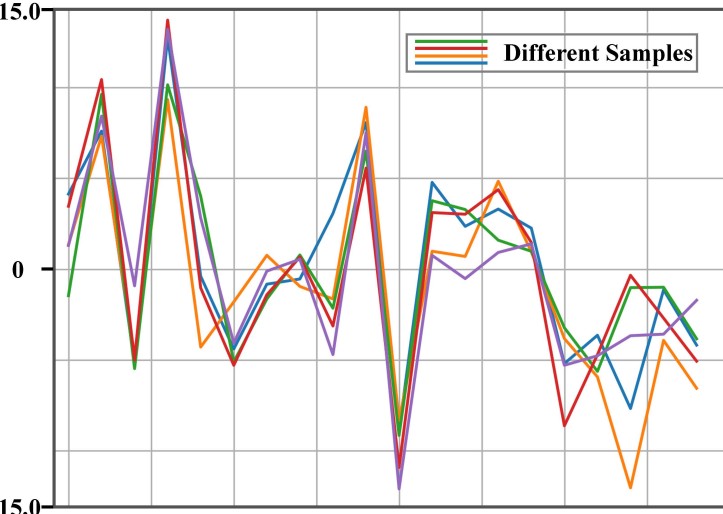

Figure A1: **Visualization of different samples' activations.** The large peaks may dominate the similarity index as their numerical values substantially surpass smaller values.

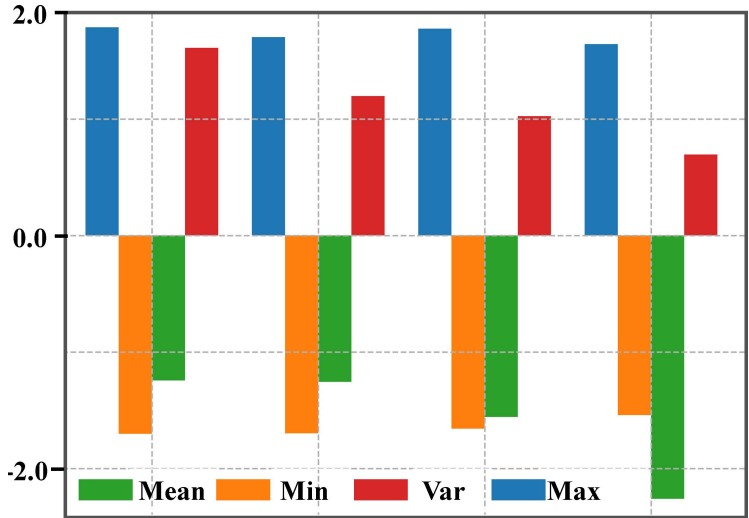

Figure A2: **Statistics of different samples' activations.** There are clear margins between different statistics (*e.g.,* Max and Min) of each sample, suggesting that the activation distribution is very peaky.

## C  User Preference Study

Here we present more details about our human perceptual judgment. Recall that two strategies are designed for different investigations, namely benchmarking the synthesis quality of one specific generative model and comparing two paired generative models. Fig. A5 shows the user interface for benchmarking the synthesis quality of one specific generative model (*i.e.,* BigGAN on ImageNet here). To be more specific, considerable randomly generated images are shown to the user, and the user is required to determine the fidelity of synthesized images. We then obtain the final scores by averaging the judgments of the participants (*i.e.,* 100 individuals).

Fig. A6 and Fig. A7 show the human evaluation results on FFHQ and ImageNet dataset, respectively. The percentages denote how many samples of the selected images are considered photo-realistic. Together with the quantitative results in our main paper, we could tell that the proposed metric shows a better correlation with human visual comparison.

Recall that in our main paper, we find that our evaluation system gives the opposite ranking to the existing metric (*i.e.,* FID) in some circumstances. For instance, the synthesis quality of ICGAN is determined basically the same as that of the class-conditional ICGAN (C-ICGAN) under our evaluation, whereas the FID votes C-ICGAN for the much better one. We thus conduct the other user

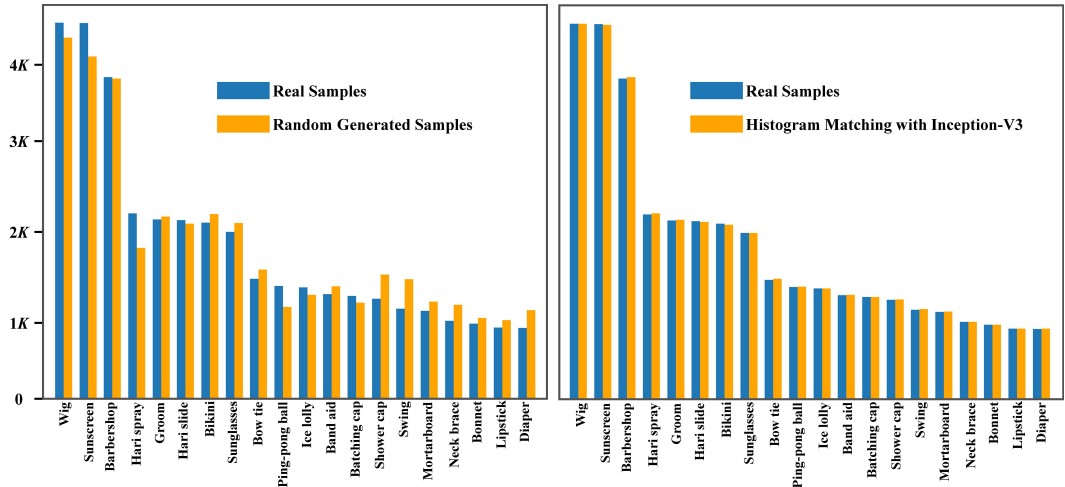

Figure A3: **The class distribution of randomly generated images (*left*)** and **histogram matched images (*right*),** predicted by the fully-supervised Inception-V3 [56].

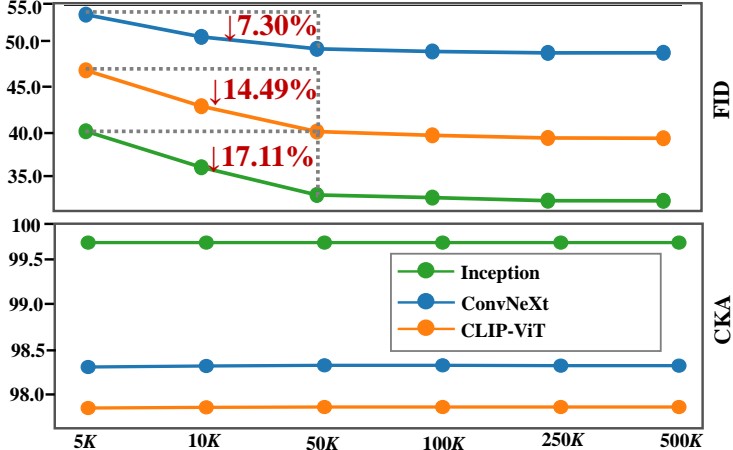

Figure A4: **Fréchet Distance (FD) and Centered Kernel Alignment (CKA) scores evaluated under various data regimes on ImageNet dataset.** FID scores are scaled for better visualization. ↓ denotes the results fluctuate downward. The percentages represent the magnitude of the numerical variation.

study to compare two paired generative models. Concretely, we prepare groups of paired images of different generative models and ask 100 individuals to assess which model could produce high-quality images. The same groups are repeated several times by changing the order of images, ensuring the human evaluation is reliable and consistent.

 Fig. A8 provides the interface of comparing two paired generative models, users are asked to choose which set of images looks more plausible. Additionally, Fig. A9 shows the pipeline of analyzing the paired comparison results. Specifically, the same groups of images are repeated for 4 times in random order and users are shown 16 images from two models to determine the more photorealistic one. In this way, the results of choosing both Projected-GAN and StyleGAN2 two times are identified as indistinguishable for enduring the consistency. Namely, the users choose different rankings between the two sets when the order of images is changed, which does not meet the consistency. Consequently, the final scores for paired comparison are obtained by quantifying the percentage of the human preferences that correlate the consistency.

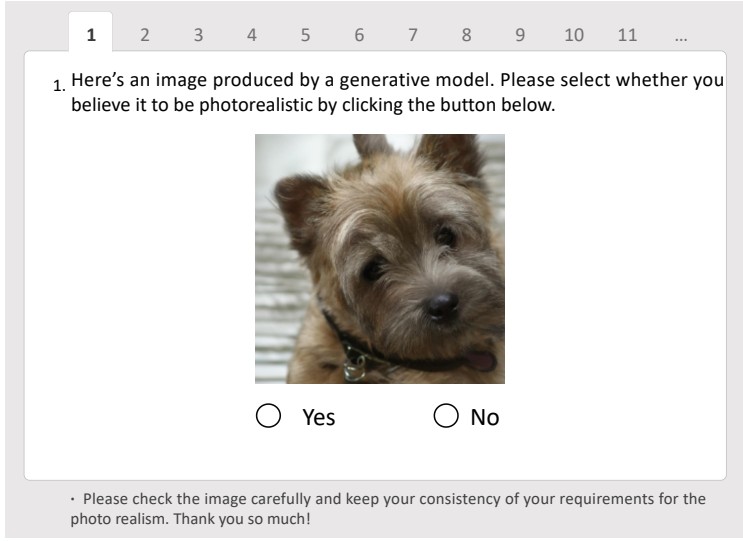

Figure A5: **User interface for benchmarking the synthesis quality.**

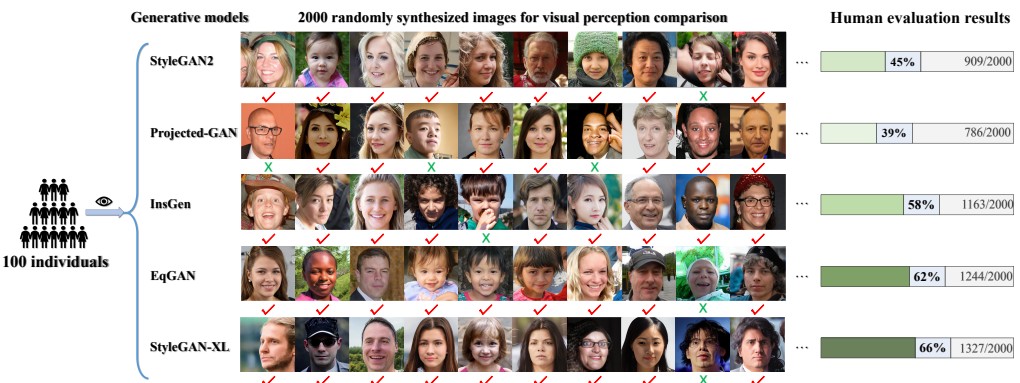

Figure A6: **Human judgment results of various generative models on FFHQ.** $2K$ images randomly generated by different models are selected for comparison.

## D  More Quantitative and Qualitative Results

In this section, we provide more quantitative and qualitative results to further demonstrate the efficacy of our newly developed measurement system.

**Comparing with more metrics.** In order to further investigate the efficacy of our proposed metric, we we further involve Kernel Inception Distance (KID) [4], precision, and recall [49] into our comparison. Note that the original KID employs Inception-V3 as the feature extractor, and there is a large "perceptual null space" in Inception-V3. Therefore, we first investigate whether KID scores can be altered by attacking the feature extractor with the histogram matching mechanism. The experimental details are consistent with computing Fréchet Distance (FD$_\downarrow$) in Tab.2 of the main paper. Tab. A4 presents the quantitative results. Still, some extractors, such as Inception, Swin-Transformer, and ResMLP, are susceptible to the histogram matching attack. For instance, the KID score of Swin-Transformer is improved by 5.31% when the chosen set is used. These observations agree with our findings in our main paper, suggesting that certain extractors can be hacked when KID is employed as the distributional distance. Then, we investigate the sample efficiency of KID, Precision, and Recall to probe the impacts of the amount of generated samples. Fig. A10 presents the curves of KID, Precision, and Recall scores computed under different data regimes. Similarly, we could observe that the KID scores can be improved by synthesizing more images. Interestingly, the recall scores decrease as the generated sample size increases whereas the precision is stable. This is caused by the definition of recall: recall measures the proportion of the real distribution that is covered by

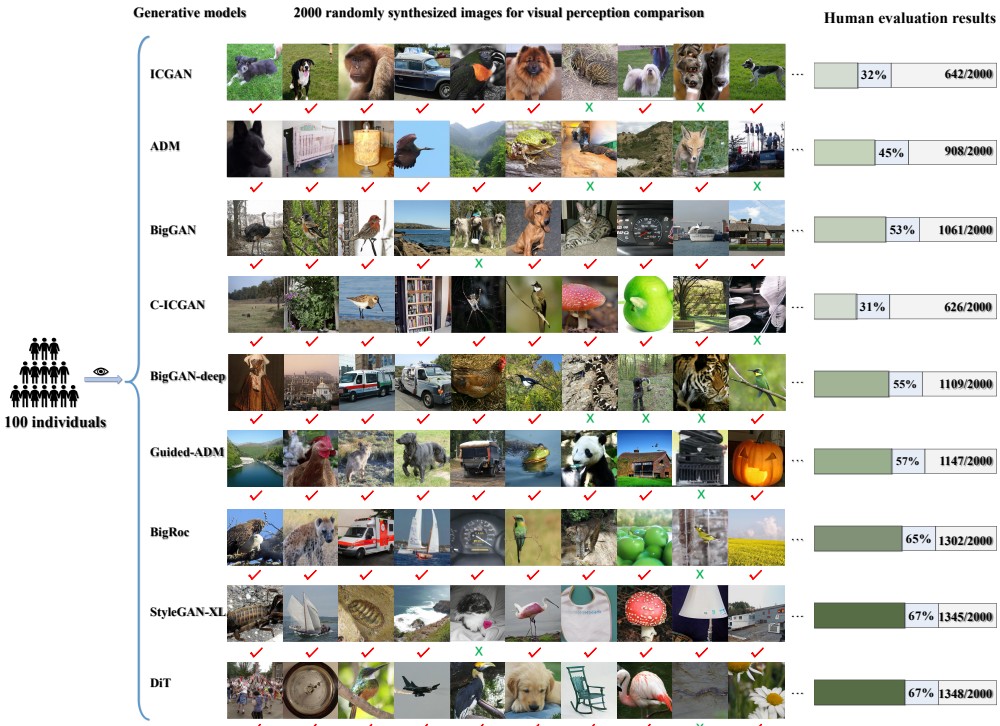

Figure A7: **Human judgment results of various generative models on ImageNet.** $2K$ images randomly generated by different models are selected for comparison.

Table A4: **Quantitative comparison results of Kernel Inception Distance (KID$_\downarrow$, $\times e^{-3}$) on FFHQ dataset**. "Random, Chosen$_I$" respectively represent the synthesized distribution of randomly generated and matching the class prediction of Inception-V3. Moreover, "$_R$" and "$_V$" respectively denote the architecture of ResNet and ViT. ($\downarrow$) indicates the results are hacked by the histogram matching mechanism. Notably, the values across different rows are not comparable and we report the stds of three testings to better illustrate the numerical fluctuation of various extractors towards the histogram attack.

| Model | Inception | ConvNeXt | SWAV | MoCo-R | RepVGG | CLIP-R |
|---|---|---|---|---|---|---|
| Random | $1.88_{\pm0.02}$ | $34.81_{\pm0.11}$ | $9.61_{\pm0.06}$ | $5.31_{\pm0.06}$ | $33.88_{\pm0.29}$ | $2.85_{\pm0.05}$ |
| Chosen$_I$ | $1.71_{\pm0.02\downarrow}$ | $34.82_{\pm0.10}$ | $9.61_{\pm0.06}$ | $5.31_{\pm0.05}$ | $33.89_{\pm0.27}$ | $2.85_{\pm0.05}$ |

| Model | Swin | ViT | DeiT | CLIP-V | MoCo-V | ResMLP |
|---|---|---|---|---|---|---|
| Random | $21.64_{\pm0.10}$ | $16.74_{\pm0.10}$ | $18.01_{\pm0.19}$ | $38.06_{\pm0.20}$ | $15.41_{\pm0.09}$ | $4.86_{\pm0.02}$ |
| Chosen$_I$ | $20.49_{\pm0.09\downarrow}$ | $16.74_{\pm0.12}$ | $19.39_{\pm0.22}$ | $38.09_{\pm0.19}$ | $15.40_{\pm0.07}$ | $4.70_{\pm0.02\downarrow}$ |

the synthesized distribution. In practical computation, the denominator increases as the synthesized samples increases, while the numerator (*i.e.,* images from the real distribution) remain unchanged. In this way, the recall scores decrease as the generated sample size increases and vice versa. By contrast, CKA scores are stable under different data regimes, (please see Fig. 2 in the main paper). Moreover, CKA can provide reliable synthesis evaluation that agrees with human visual judgment. Accordingly, CKA is a proper choice for building a consistent and reliable measurement system.

**More hackability results of Fréchet Distance on ImageNet.** In addition to evaluating the robustness of these extractors on the FFHQ dataset, we further perform the same experiment on the ImageNet dataset. Tab. A5 presents the quantitative results. We can tell from these results that the chosen feature extractors are robust to the attack, further demonstrating their reliability.

**More results of MLP-based extractors.** We further incorporate two MLP-based models as the feature extractor for synthesis evaluation, namely namely gMLP [37] and MLP-mixer [59]. Following

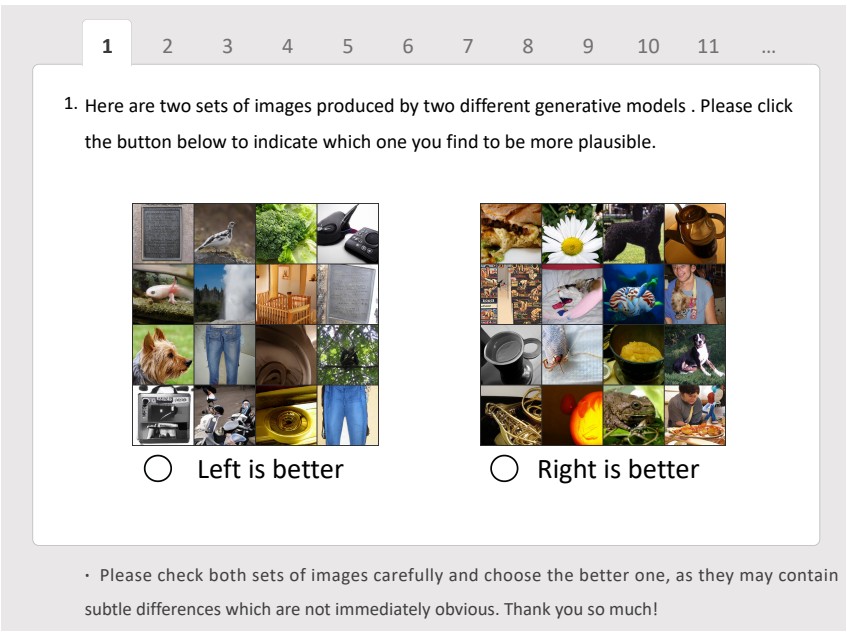

Figure A8: **User interface for comparing the synthesis quality of two paired generative models.** People are asked to determine which set of images look more photorealistic.

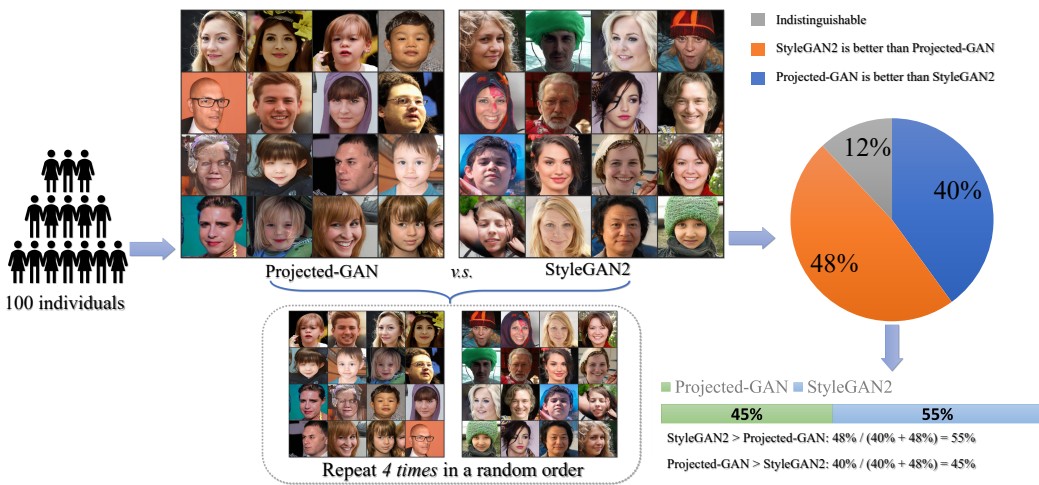

Figure A9: **The pipeline of analyzing the paired comparison results.**

the experimental settings in our main paper, we identify the reliability and robustness of these MLP-based models via 1) visualizing the highlighted regions that contribute most significantly to the measurement results, and 2) attacking the feature extractor with histogram matching attack. Tab. A6 presents the qualitative (*left*) and quantitative (*right*) results. On one hand, the heatmap visualization results indicate that both gMLP and mixer-MLP capture limited semantics. Considering that more visual semantics should be considered for a more comprehensive evaluation, gMLP and MLP-mixer might not be adequate for synthesis comparison. On the other hand, the quantitative results demonstrate that their FD scores could be altered by the histogram matching attack, without actually improving the synthesis quality. That is, gMLP and MLP-mixer are susceptible to the histogram attack. Together with the finding that the FD scores of ResMLP could be manipulated without any improvement to the generative models in Tab. 2 of our main paper, we do not integrate MLP-based feature extractors into our measurement system.

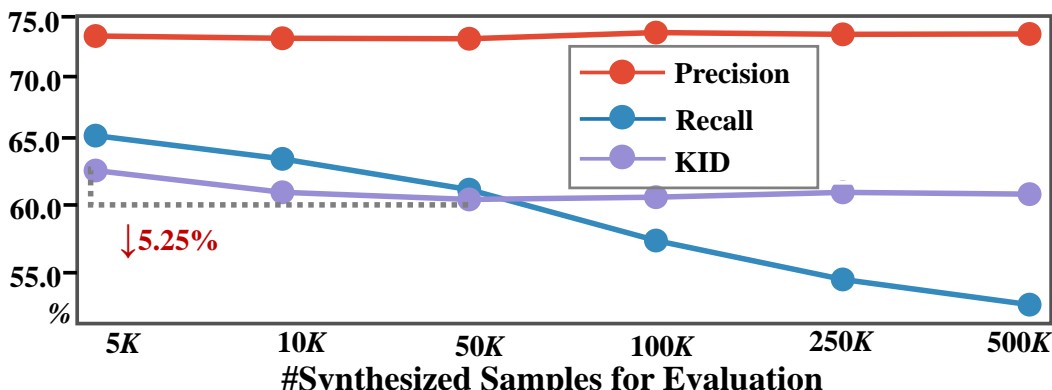

Figure A10: **Kernel Inception Distance (KID), Precision, and Recall scores evaluated under various data regimes on FFHQ dataset.** The scores are scaled for better visualization. ↓ denotes the results fluctuate downward. The percentages represent the magnitude of the numerical variation.

Table A5: **Quantitative comparison results of Fréchet Distance (FD$_\downarrow$) on ImageNet dataset.** "Random, Chosen$_I$" respectively represent the synthesized distribution of randomly generated and matching the class prediction of Inception-V3. Moreover, "$_R$" and "$_V$" respectively denote the architecture of ResNet and ViT. ($_\downarrow$) indicates the results are hacked by the histogram matching mechanism. Notably, the values across different rows are not comparable and we report the stds of three testings to better illustrate the numerical fluctuation of various extractors towards the histogram attack.

| Model | Inception | ConvNeXt | SWAV | MoCo-R | RepVGG | CLIP-R |
|---|---|---|---|---|---|---|
| Random | $34.29_{\pm0.09}$ | $78.02_{\pm0.16}$ | $0.13_{\pm0.003}$ | $0.32_{\pm0.002}$ | $54.98_{\pm0.22}$ | $27.64_{\pm0.15}$ |
| Chosen$_I$ | $33.05_{\pm0.08\downarrow}$ | $78.10_{\pm0.14}$ | $0.13_{\pm0.002}$ | $0.32_{\pm0.002}$ | $54.30_{\pm0.24}$ | $27.66_{\pm0.17}$ |

| Model | Swin | ViT | DeiT | CLIP-V | MoCo-V | ResMLP |
|---|---|---|---|---|---|---|
| Random | $323.12_{\pm0.88}$ | $50.97_{\pm0.20}$ | $621.98_{\pm1.02}$ | $5.46_{\pm0.09}$ | $50.01_{\pm0.21}$ | $145.32_{\pm1.02}$ |
| Chosen$_I$ | $301.91_{\pm0.92\downarrow}$ | $50.96_{\pm0.18}$ | $597.32_{\pm1.11\downarrow}$ | $5.46_{\pm0.07}$ | $50.00_{\pm0.19}$ | $133.06_{\pm1.09\downarrow}$ |

Table A6: **Heatmaps from MLP-based extractors, namely Mixer-MLP [59] and gMLP [37].** (*left*) and **Quantitative comparison results of MLP-based extractors' Fréchet Distance (FD$_\downarrow$) on the ImageNet dataset (*right*)** . ViT [16] serves as the feature extractor for hierarchical evaluation here.

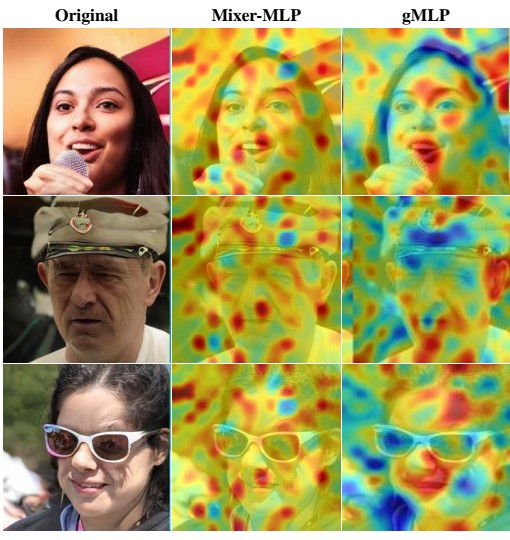

| Extractor | Random | Chosen$_I$ |
|---|---|---|
| gMLP | $2.93_{\pm0.004}$ | $2.89_{\pm0.004\downarrow}$ |
| mixer-MLP | $5.51_{\pm0.01}$ | $5.35_{\pm0.01\downarrow}$ |

**Similarities between various representation spaces.** Recall that we filtered out extractors that define similar representation spaces to avoid redundancy in the main paper. The correlation between

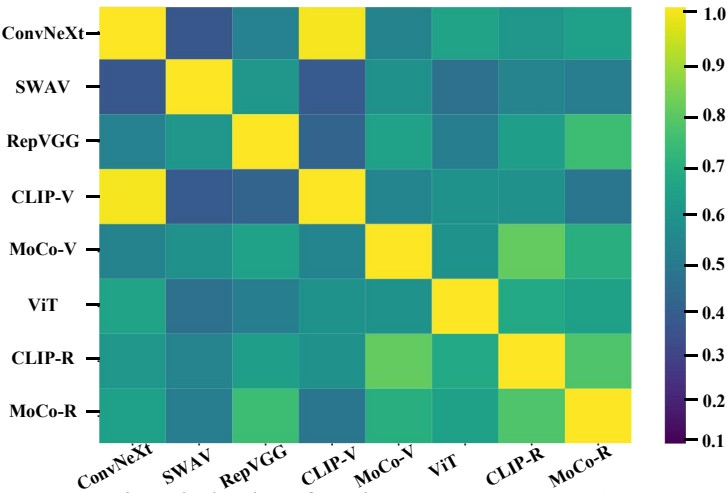

Figure A11: **Representation similarity of various extractors.** Darker Yellow denotes higher similarity.

representations of high dimension in different feature extractors is calculated following [32]. In particular, a considerable number of images (*i.e.,* $10K$ images from ImageNet) are fed into these extractors for computing their correspondence. Fig. A11 shows the similarity of their representations. Obviously, the representations of CLIP-ResNet and MoCo-ResNet have higher similarity with other extractors. Considering these two extractors are both CNN-based and they capture similar semantics with other CNN-based extractors, we remove the CLIP-ResNet and MoCo-ResNet to avoid redundancy. Accordingly, we obtain a set of feature extractors that 1) capture rich semantics in a complementary way, 2) are robust toward the histogram matching attack, and 3) define meaningful and distinctive representation spaces for synthesis comparison. The following table presents these feature extractors. These extractors, including both CNN-based and ViT-based architectures,

| | |
|---|---|
| **CNN-based** | ConvNeXt [40], SWAV [6], RepVGG [15] |
| **ViT-based** | CLIP-ViT [46], MoCo-ViT [9], ViT [16] |

have demonstrated strong performance in pre-defined and downstream tasks, facilitating more comprehensive and reliable evaluation. Notably, the inclusion of self-supervised extractors SWAV, CLIP-V, and MoCo-V aligns with previous findings [41, 35, 3]. This selection of feature extractors provides a diverse and complementary set of representations, enabling a more comprehensive and reliable evaluation of synthesis quality in generative models.

**More results of hierarchical levels from various extractors.** Tab. A7, Tab. A8, Tab. A9, Tab. A10, and Tab. A11 respectively present the heatmaps and quantitative results of various semantic levels. We could tell that despite the Fréchet Distance (FD) scores consistently reflect synthesis quality, their numerical values fluctuate dramatically. On the contrary, CKA provides normalized distances *w.r.t* the numerical scale across various levels. Also, the heatmaps from various semantic levels reveal that hierarchical features encode different semantics. Such observation provides interesting insights that feature hierarchy should be also considered for synthesis comparison. Notably, benefiting from the bounded quantitative results, CKA demonstrates great potentials for comparison across hierarchical layers.

Table A7: **Heatmaps from various semantic levels on FFHQ dataset (*left*) and quantitative results of Fréchet Distance (FD ↓) and Centered Kernel Alignment (CKA ↑) on ImageNet dataset (*right*)** . ConvNext [40] serves as the feature extractor for hierarchical evaluation here.

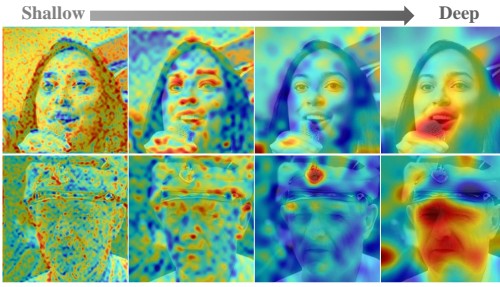

| Model | BigGAN | | BigGAN-deep | | StyleGAN-XL | |
|---|---|---|---|---|---|---|
| Layer | FD↓ | CKA↑ | FD↓ | CKA↑ | FD↓ | CKA↑ |
| Layer$_1$ | 2.64 | 96.08 | 2.56 | 96.35 | 0.58 | 98.24 |
| Layer$_2$ | 40.20 | - | 32.32 | - | 11.84 | - |
| Layer$_3$ | 687.40 | 58.76 | 364.95 | 60.25 | 264.87 | 62.53 |
| Layer$_4$ | 140.04 | 68.86 | 102.26 | 69.27 | 19.22 | 70.52 |
| Overall | N/A | 74.57 | N/A | 75.29 | N/A | 77.10 |

Table A8: **Heatmaps from various semantic levels on FFHQ dataset (*left*) and quantitative results of Fréchet Distance (FD ↓) and Centered Kernel Alignment (CKA ↑) on ImageNet dataset (*right*)** . RepVGG [15] serves as the feature extractor for hierarchical evaluation here.

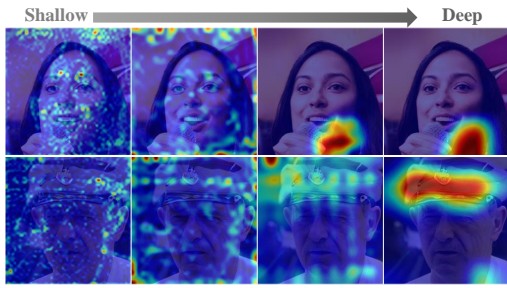

| Model | BigGAN | | BigGAN-deep | | StyleGAN-XL | |
|---|---|---|---|---|---|---|
| Layer | FD↓ | CKA↑ | FD↓ | CKA↑ | FD↓ | CKA↑ |
| Layer$_1$ | 0.35 | 96.92 | 0.32 | 97.51 | 0.04 | 98.79 |
| Layer$_2$ | 0.35 | 96.19 | 0.33 | 96.32 | 0.03 | 98.32 |
| Layer$_3$ | 0.23 | 90.05 | 0.18 | 91.15 | 0.04 | 93.51 |
| Layer$_4$ | 67.53 | 74.40 | 58.85 | 76.68 | 15.93 | 80.28 |
| Overall | N/A | 89.39 | N/A | 90.42 | N/A | 92.73 |

Table A9: **Heatmaps from various semantic levels on FFHQ dataset (*left*) and quantitative results of Fréchet Distance (FD ↓) and Centered Kernel Alignment (CKA ↑) on ImageNet dataset (*right*)** . SWAV [6] serves as the feature extractor for hierarchical evaluation here.

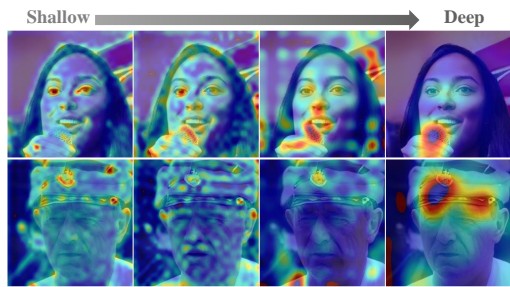

| Model | BigGAN | | BigGAN-deep | | StyleGAN-XL | |
|---|---|---|---|---|---|---|
| Layer | FD↓ | CKA↑ | FD↓ | CKA↑ | FD↓ | CKA↑ |
| Layer$_1$ | 0.67 | 99.90 | 0.46 | 99.91 | 0.07 | 99.99 |
| Layer$_2$ | 0.87 | 97.89 | 0.60 | 98.87 | 0.31 | 99.51 |
| Layer$_3$ | 16.15 | 95.60 | 12.02 | 96.21 | 1.90 | 98.15 |
| Layer$_4$ | 11.18 | 86.10 | 8.69 | 87.71 | 1.85 | 92.54 |
| Overall | N/A | 94.87 | N/A | 95.68 | N/A | 97.55 |

Table A10: **Heatmaps from various semantic levels on FFHQ dataset (*left*) and quantitative results of Fréchet Distance (FD ↓) and Centered Kernel Alignment (CKA ↑) on ImageNet dataset (*right*)** . ViT [16] serves as the feature extractor for hierarchical evaluation here.

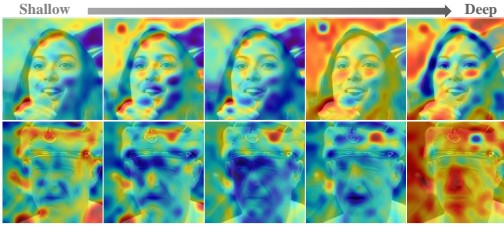

| Model | BigGAN | | BigGAN-deep | | StyleGAN-XL | |
|---|---|---|---|---|---|---|
| Layer | FD↓ | CKA↑ | FD↓ | CKA↑ | FD↓ | CKA↑ |
| Layer$_1$ | 0.20 | 99.62 | 0.19 | 99.67 | 0.01 | 99.97 |
| Layer$_2$ | 1.31 | 97.75 | 1.19 | 97.92 | 0.18 | 99.76 |
| Layer$_3$ | 6.93 | 97.53 | 6.06 | 97.63 | 1.22 | 99.67 |
| Layer$_4$ | 29.95 | 96.49 | 23.98 | 97.20 | 8.51 | 98.72 |
| Overall | N/A | 97.85 | N/A | 98.11 | N/A | 99.53 |

Table A11: **Heatmaps from various semantic levels on FFHQ dataset (*left*) and quantitative results of Fréchet Distance (FD ↓) and Centered Kernel Alignment (CKA ↑) on ImageNet dataset (*right*)** . MoCo-ViT [9] serves as the feature extractor for hierarchical evaluation here.

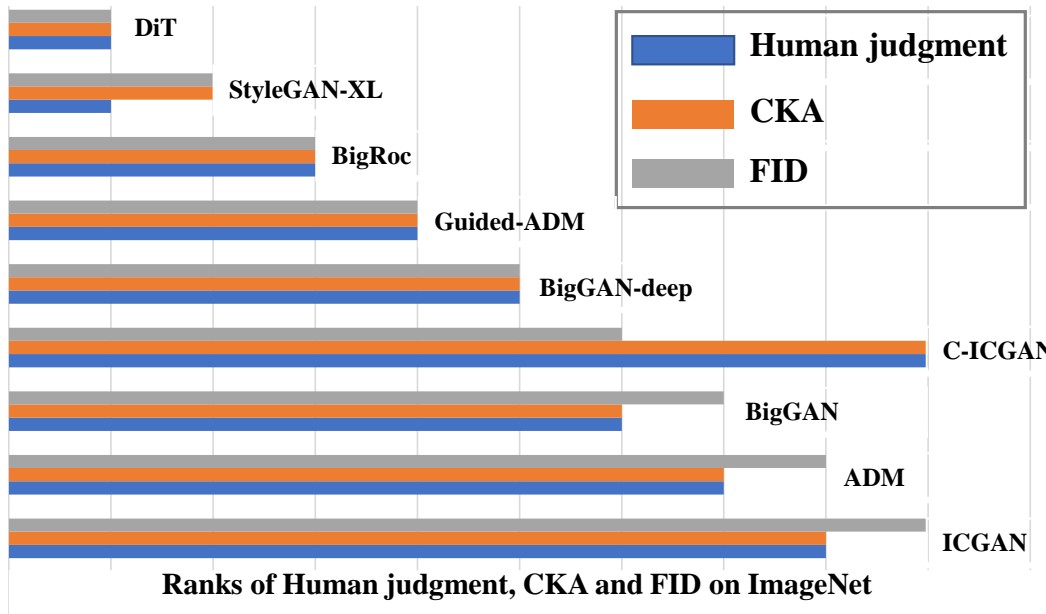

| Model | BigGAN | | BigGAN-deep | | StyleGAN-XL | |
|---|---|---|---|---|---|---|
| Layer | FD↓ | CKA↑ | FD↓ | CKA↑ | FD↓ | CKA↑ |
| Layer$_1$ | 0.10 | 98.62 | 0.05 | 99.04 | 0.04 | 99.97 |
| Layer$_2$ | 1.01 | 97.15 | 0.68 | 97.30 | 0.43 | 99.64 |
| Layer$_3$ | 9.18 | 96.07 | 9.01 | 96.77 | 4.30 | 99.11 |
| Layer$_4$ | 3.35 | 97.25 | 3.22 | 97.82 | 1.85 | 99.00 |
| Overall | N/A | 97.27 | N/A | 97.73 | N/A | 99.43 |

## E  Visualized quantitative results

To make our results easier to parse, we visualize the correlation between different metrics and the human evaluation results. Specifically, we plot the correlation of the averaged ranks of various models given by human judgment, CKA, and FID. Fig. A12 and Fig. A13 respectively present the visualization results of the ImageNet, FFHQ, and LSUN-Church datasets. Obviously, the averaged ranks given by CKA are more consistent with that of the human evaluation, demonstrating the accuracy of CKA. Moreover, we plot the comparison between the stds and the improvements obtained by the histogram attack for better illustration. Fig. A14 presents the results. Similarly, we could observe that the improvement is actually caused by the histogram attack rather than the variance of attempts.

Figure A12: **The correlation of the averaged ranks of various models on ImageNet given by human judgment, CKA, and FID.**

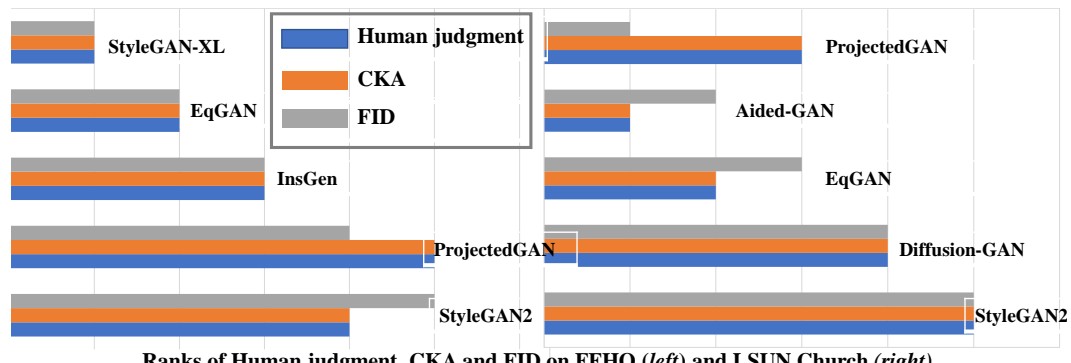

Ranks of Human judgment, CKA and FID on FFHQ (*left*) and LSUN Church (*right*)

Figure A13: **The correlation of the averaged ranks of various models on FFHQ and LSUN-Church given by human judgment, CKA, and FID.**

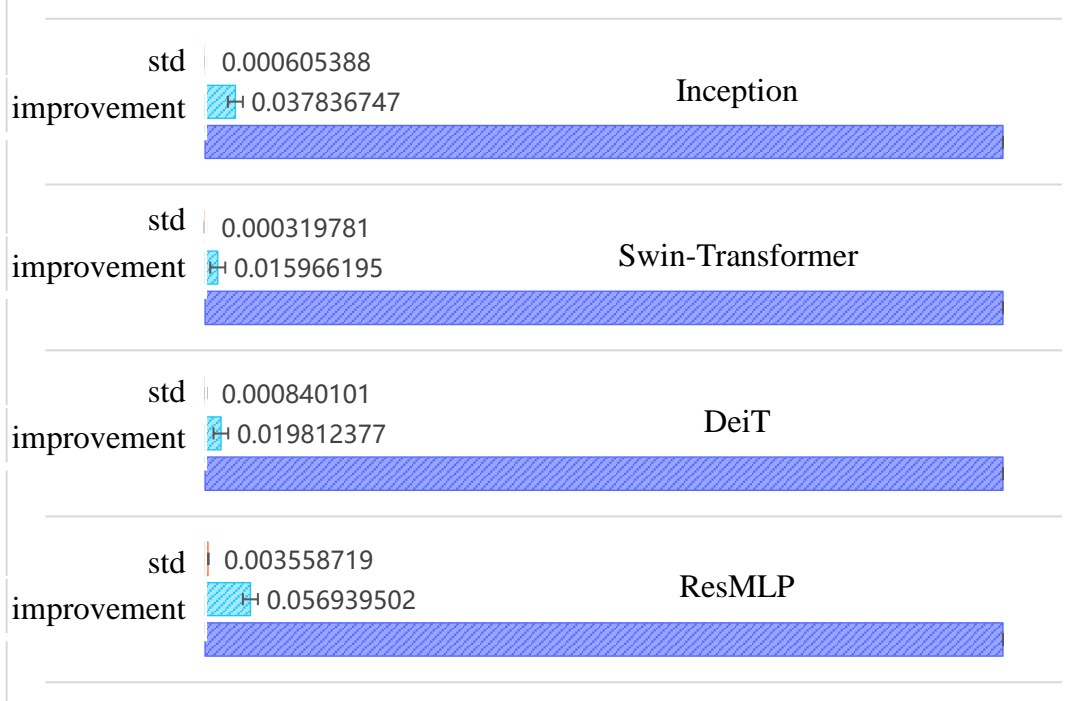

Figure A14: **The quantitative comparison between the stds and the improvements obtained by the histogram attack.**

