# Rebuttal for " Revisiting the Evaluation of Image Synthesis with GANs "

1  **To Reviewer cWrw**

2  **Q1: Include state-of-the-art generative models like diffusion models.**

3  **Reply**: Thanks. Following your valuable suggestion, we further include more state-of-the-
4  art generative models on the ImageNet dataset for synthesis evaluation, namely GigaGAN
5  (CVPR'2023) [7], MDT (ICCV'2023) [4], and DG-Diffusion (ICML'2023) [8]. Specifically, we
6  either gather their official models for inference or download the pre-generated images released by
7  the authors for evaluation. Similarly, $50K$ generated images and the entire training set (*i.e.,* $1.28M$
8  images) are used as the synthesized and real distributions, respectively. All details are consistent with
9  the experiments conducted in our main paper. Tab. 1 presents the quantitative comparison results.
10  Akin to the results in our main paper, our evaluation system provides consistent ranks with FID and
11  human visual evaluation, demonstrating the reliability of our metric. These results will be added in
12  the next version of our paper.

Table 1: **Quantitative comparison results of Centered Kernel Alignment (CKA$_\uparrow$) on ImageNet dataset**. $^\dagger$ scores are quoted from the original paper and others are tested three times.

| Model | FID$^\dagger$ | ConvNeXt | RepVGG | SWAV | ViT | MoCo-ViT | CLIP-ViT | Overall | User study |
|---|---|---|---|---|---|---|---|---|---|
| GigaGAN [7] | 3.45 | 68.01 | 79.93 | 90.15 | 98.34 | 82.40 | 96.52 | 85.89 | 65% |
| DG-Diffusion [8] | 3.18 | 68.22 | 80.06 | 90.56 | 98.46 | 82.51 | 96.88 | 86.12 | 66% |
| MDT [4] | 1.79 | 69.64 | 81.68 | 91.78 | 99.43 | 83.43 | 98.19 | 87.36 | 69% |

13  **Q2: Provide the evaluation on more MLP-based models like mlp-mixer.**

14  **Reply**: Thanks. As suggested, two MLP-based models are leveraged as the feature extractor for
15  synthesis evaluation, namely gMLP [12] and MLP-mixer [20]. Following the experimental settings
16  in our main paper, we identify the reliability and robustness of these MLP-based models via 1)
17  visualizing the highlighted regions that contribute most significantly to the measurement results, and
18  2) attacking the feature extractor with histogram matching attack. Fig. 1 and Tab. 2 respectively
19  present the qualitative and quantitative results. On one hand, the heatmap visualization results indicate
20  that both gMLP and mixer-MLP capture limited semantics. Considering that more visual semantics
21  should be considered for a more comprehensive evaluation, gMLP and MLP-mixer might not be
22  adequate for synthesis comparison. On the other hand, the quantitative results demonstrate that
23  their FD scores could be altered by the histogram matching attack, without actually improving the
24  synthesis quality. That is, gMLP and MLP-mixer are susceptible to the histogram attack. Together
25  with the finding that the FD scores of ResMLP could be manipulated without any improvement to the
26  generative models in Tab.2 of our main paper, we do not integrate MLP-based feature extractors into
27  our measurement system. These results will be added in the next version of our paper.

28  **Q3: 100 Human Judgment may not enough to fully capture the complexities of evaluating**
29  **generative models objectively.**

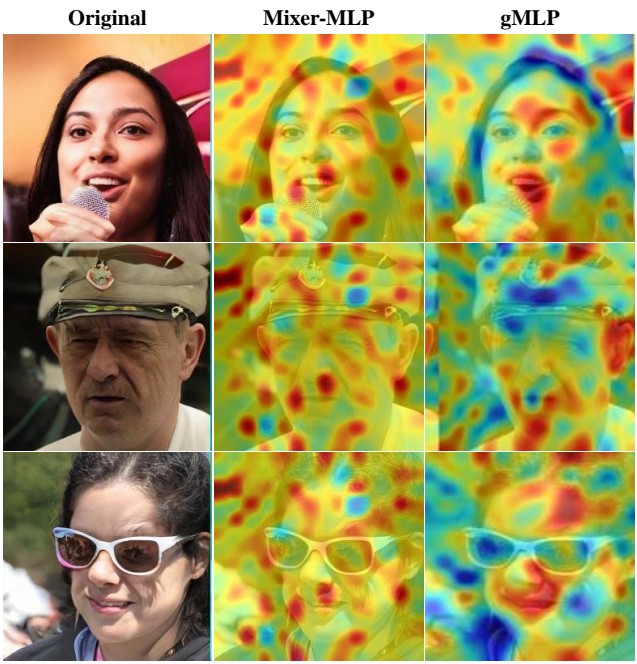

| Original | Mixer-MLP | gMLP |

Figure 1: **Heatmaps from MLP-based extractors, namely Mixer-MLP [20] and gMLP [12].**

Table 2: **Quantitative comparison results of MLP-based extractors' Fréchet Distance ($FD_\downarrow$) on the ImageNet dataset**. [†] scores are quoted from the original paper and others are tested three times.

| Extractor | Random | $Chosen_I$ |
|---|---|---|
| gMLP | $2.93_{\pm0.004}$ | $2.89_{\pm0.004\downarrow}$ |
| mixer-MLP | $5.51_{\pm0.01}$ | $5.35_{\pm0.01\downarrow}$ |

**Reply**: Thanks. We agree that involving thousands of persons for human visual evaluation can provide more consistent and reliable results. However, this is too expensive for us as including thousands of participants requires massive human and time resources. Therefore, two strategies of human perceptual judgment are designed for different investigations in our main experiments, namely benchmaking the synthesis quality of one specific generative model and comparing two paired models. In particular, 100 participants are asked to vote the synthesis quality and their final scores are averaged to avoid overly subjective individual outcomes. Moreover, in order to ensure that our human evaluation is reliable and consistent, we repeat the same images several times (*i.e.,* 4) randomly for human visual comparison. In this way, if one user vote photorealistic and unrealistic two times each for the same images, the results would be considered as indistinguishable. This operation further filters overly subjective individual judgment and ensure the rationality of our user study. Additionally, we notice that common choice for human evaluation in the community is to include about 50 participants for perceptual comparison [18, 23, 14, 22, 16, 11, 10]. For instance, [23], [18] and [10] asked 50 workers to pick the unrealistic images, ProjectedGAN [16] conducted a human preference study with only 28 participants, and the most recent work [14] included only 15 graders to compare the synthesis quality. By contrast, 100 persons are involved in our human judgment, thus we believe that our perceptual comparison results are reliable. Furthermore, we view large-scale human evaluation as our future work to perform more extensive investigations.

Hope that the above discussions could address your concerns, please let us know if you have any further questions. Thanks for your effort and constructive suggestions again.

**To Reviewer Z9sB**

**Q1: Only the mean values of metrics are reported, no stds.**

**Reply**: Thanks. As suggested, we add the std values of our experiments to better illustrate the numerical fluctuation of various extractors towards the histogram attack. 3 presents the quantitative

Table 3: **Quantitative comparison results of Fréchet Distance (FD$_\downarrow$) on FFHQ dataset**. "Random, Chosen$_I$" respectively represent the synthesized distribution of randomly generated and matching the class prediction of Inception-V3. Moreover, "$_v$" and "$_v$" respectively denote the architecture of ResNet and ViT. ($_\downarrow$) indicates the results are hacked by the histogram matching mechanism. Notably, the values across different rows are not comparable and the results are tested three times.

| Model | Inception | ConvNeXt | SWAV | MoCo$_r$ | RepVGG | CLIP$_r$ | Swin | ViT | DeiT | CLIP$_v$ | MoCo$_v$ | ResMLP |
|---|---|---|---|---|---|---|---|---|---|---|---|---|
| Random | 2.81$_{\pm0.01}$ | 78.03$_{\pm0.10}$ | 0.13$_{\pm0.002}$ | 0.24$_{\pm0.003}$ | 129.61$_{\pm0.41}$ | 10.34$_{\pm0.06}$ | 142.87$_{\pm0.12}$ | 15.11$_{\pm0.09}$ | 437.80$_{\pm0.14}$ | 1.06$_{\pm0.01}$ | 7.32$_{\pm0.03}$ | 99.11$_{\pm0.06}$ |
| Chosen$_I$ | 2.65$_{\pm0.01\downarrow}$ | 78.19$_{\pm0.11}$ | 0.13$_{\pm0.002}$ | 0.24$_{\pm0.003}$ | 129.67$_{\pm0.39}$ | 10.36$_{\pm0.08}$ | 140.01$_{\pm0.12\downarrow}$ | 15.11$_{\pm0.10}$ | 430.81$_{\pm0.16\downarrow}$ | 1.06$_{\pm0.01}$ | 7.40$_{\pm0.03}$ | 95.36$_{\pm0.06\downarrow}$ |

results. We could tell that the FD scores of extractors that are vulnerable to the attack can be improved by matching the histogram, and the improvement of FD scores is greater than stds. For instance, the improvement of FD scores from the Inception model is 0.16 and the computation std is only 0.01, there is an order of magnitude difference between them. Moreover, the improvement of FD scores from the Swin-Transformer model is 2.86 and the computation std is only 0.12. That is, the improvement is actually caused by the histogram attack rather than the variance of attempts. Note that the generator is unchanged but the FD scores are improved by the attack, which is unacceptable for synthesis evaluation. Accordingly, extractors that are vulnerable to the histogram matching attack are not reliable for evaluation.

**Q2: The authors provide many tables with the results but it is not trivial to parse them. Specifically, checking whether this or that metric correlates with the human evaluation ahowld be done manually. It would be great if this could be somehow quantified or visualized (e.g., FID/other metrics as functions of the user score, 2D plots).**

**Reply**: Thanks. Following your valuable suggestion, we visualize the correlation between different metrics and the human evaluation results. Specifically, we plot the correlation of the averaged ranks of various models given by human judgment, CKA, and FID. Fig. 2 and Fig. 3 respectively present the visualization results of the ImageNet, FFHQ, and LSUN-Church datasets. Obviously, the averaged ranks given by CKA are more consistent with that of the human evaluation, demonstrating the accuracy of CKA. Moreover, we plot the comparison between the stds and the improvements obtained by the histogram attack for better illustration. Fig. 4 presents the results. Similarly, we could observe that the improvement is actually caused by the histogram attack rather than the variance of attempts.

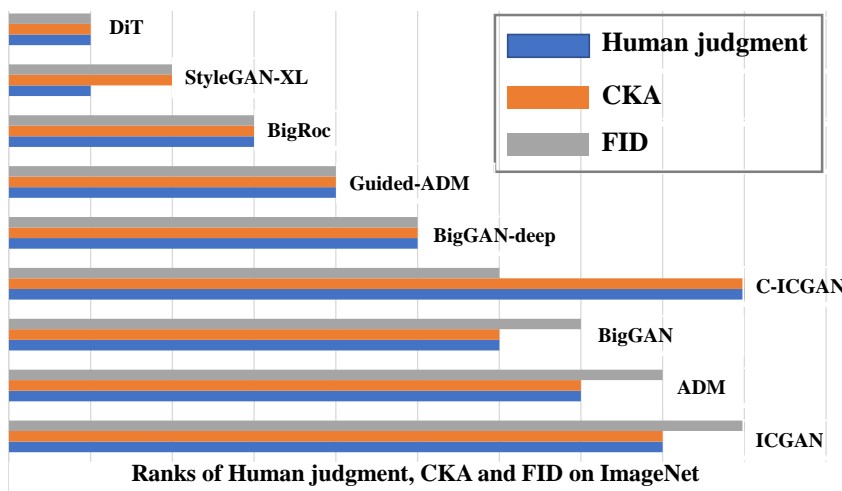

Figure 2: **The correlation of the averaged ranks of various models on ImageNet given by human judgment, CKA, and FID.**

**Q3: May be it is more fair to emphasize other advantages of CKA (such as the sample efficiency) rather than consistency and reliability.**

**Reply**: Thanks. On one hand, our results demonstrate that CKA provides a consistent ranking with the FID scores in most cases, demonstrating that CKA can deliver the similarity between different data distributions. One the other hand, CKA agrees with human visual judgment whereas FID fails in

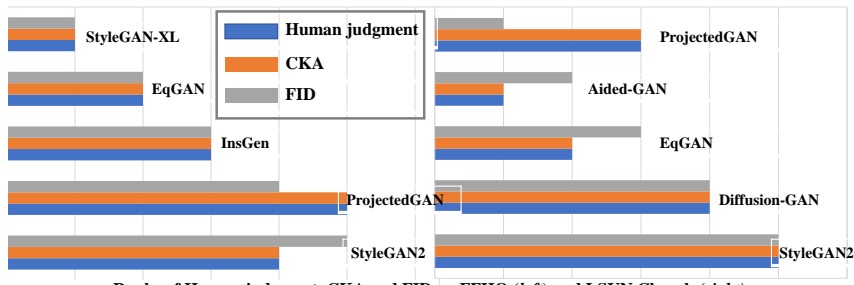

Ranks of Human judgment, CKA and FID on FFHQ (*left*) and LSUN Church (*right*)

Figure 3: **The correlation of the averaged ranks of various models on FFHQ and LSUN-Church given by human judgment, CKA, and FID.**

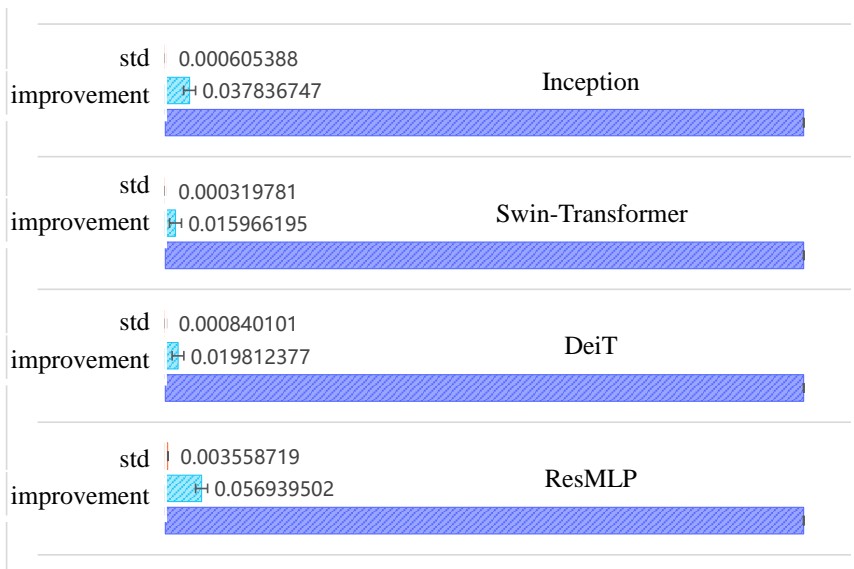

Figure 4: **The quantitative comparison between the stds and the improvements obtained by the histogram attack.**

some circumstances. That is, CKA can measure the synthesis performance more reliable than FID. Additionally, CKA shows better sample efficiency than both FID and KID. Thus we integrate CKA as the distributional distance to evaluate the synthesis performance in our system. Together with several robust feature extractors, our new measurement system is more consistent and reliable than exiting alternatives. In the main paper, we emphasize the reliability and consistency of our entire system rather than only the distributional distance (*i.e.,* CKA) as both the extractors and distances are important. We will proofread our presentation and emphasize the advantages of our overall system following your valuable suggestions.

**Q4: Include at least some evaluation/comparison/comment with this KID (both in terms of correlation with human evaluation and sample efficiency).**

**Reply**: Thanks. Following your valuable suggestion, we further involve Kernel Inception Distance (KID) [1], precision, and recall [15] into our comparison. Note that the original KID employs Inception-V3 as the feature extractor, and there is a large "perceptual null space" in Inception-V3. Therefore, we first investigate whether KID scores can be altered by attacking the feature extractor with the histogram matching mechanism. The experimental details are consistent with computing Fréchet Distance ($FD_\downarrow$) in Tab.2 of the main paper. Tab. 6 presents the quantitative results. Still, some extractors, such as Inception, Swin-Transformer, and ResMLP, are susceptible to the histogram matching attack. For instance, the KID score of Swin-Transformer is improved by 5.31% when the chosen set is used. These observations agree with our findings in our main paper, suggesting that certain extractors can be hacked when KID is employed as the distributional distance. Then, we investigate the sample efficiency of KID, Precision, and Recall to probe the impacts of the amount

of generated samples. Fig. 5 presents the curves of KID, Precision, and Recall scores computed under different data regimes. Similarly, we could observe that the KID scores can be improved by synthesizing more images. Interestingly, the recall scores decrease as the generated sample size increases whereas the precision is stable. This is caused by the definition of recall: recall measures the proportion of the real distribution that is covered by the synthesized distribution. In practical computation, the denominator increases as the synthesized samples increases, while the numerator (*i.e.,* images from the real distribution) remain unchanged. In this way, the recall scores decrease as the generated sample size increases and vice versa. By contrast, CKA scores are stable under different data regimes, (please see Fig. 2 in the main paper). Moreover, CKA can provide reliable synthesis evaluation that agrees with human visual judgment. Accordingly, CKA is a proper choice for building a consistent and reliable measurement system.

**Q5: The authors should better explain CKA metric in the main text.**
**Reply**: Thanks. Following your valuable suggestion, we add more details of the CKA metric as follows:

**Centered Kernel Alignment (CKA)** as a widely used similarity index for quantifying neural network representations [2, 9, 3], could also serve as a metric of similarity between two given distributions. To be specific, CKA is normalized from Hilbert-Schmidt Independence Criterion (HSIC) [5] to ensure invariant to isotropic scaling and is calculated by

$$\text{CKA}(\text{X}, \text{Y}) = \frac{\text{HSIC}(x, y)}{\sqrt{\text{HSIC}(x, x)\text{HSIC}(y, y)}}. \tag{1}$$

Here, HSIC determines whether two distributions are independent. Formally, let $K_{ij} = k\left(\text{x}_i, \text{x}_j\right)$ and $L_{ij} = l\left(\text{y}_i, \text{y}_j\right)$, where $k$ and $l$ are two kernels. HSIC is defined as

$$\text{HSIC}(K, L) = \frac{1}{(n-1)^2} \text{Tr}(KHLH), \tag{2}$$

where $H$ denotes the centering matrix (*i.e.,* $H_n = I_n - \frac{1}{n}\mathbf{1}\mathbf{1}^T$). For kernel selections of $k$ and $l$, we find that different kernels (RBF, polynomial, and linear) give similar results and rankings, and the RBF kernel contributes to the distinguishability of quantitative results. Therefore, RBF kernel is used for all experiments, and the bandwidth is set as a fraction of the median distance between examples [9]. These metrics are compared in a consistent setting for fair comparison, more implementation details are given in *Supplementary Material*.

**Q6: Provide a more explanatory discussion of what is CKA (beside the formulas) and some intuition what it measure and why it is a good metric?**
**Reply**: Thanks. As a widely used similarity index for measuring the correspondence between representations in neural networks, CKA has been identified to have several advantages: 1) CKA is invariant to orthogonal transformation and isotropic scaling, making is stable under various image transformations; 2) CKA can capture the non-linear correspondence between representations due to its kernel mapping; and 3) CKA can determine the correspondence across different features and with different widths, whereas previous metrics fail [9]. Additionally, through extensive experiments, we demonstrate that CKA can provide a accurate evaluation for synthesis comparison and is sample-efficient. Accordingly, CKA is a good metric for delivering the distributional discrepancy.

**Q7: What is the "null space" of this metric?**
**Reply**: Thanks. In fact, it is the feature extractor that might have a "perceptual null space". For instance, the Inception model has been identified to have a large "perceptual null space", leading it vulnerable to the histogram matching attack. Moreover, CKA measures the similarity between different distributions, and $\text{CKA}(\text{X}, \text{Y}) = 1$ if and only if the two sets coincide.

**Q8: What is the computational complexity of CKA compared to FID? How it scales with the feature dimension and sample size?**
**Reply**: Thanks. Assume $N$ samples from the evaluated distributions are used for calculating CKA, the main computational complexity of CKA comes from: 1) centering the kernel matrix with the pre-defined centering matrix with the complexity of $\mathcal{O}(N^2)$; and 2) computing HSIC scores with

the complexity of $\mathcal{O}(N^3)$. Therefore, the overall computational complexity of CKA is $\mathcal{O}(N^3)$. By contrast, the computational complexity of FID mainly comes from calculating the mean and variance of the sample features ($N \times d$, where $d$ denotes the feature dimension). The computational complexity is $\mathcal{O}(N \times d)^3$. The computational complexity of both CKA and FID increases linearly with the cubic power of sample size. Moreover, as suggested, we provide the clock time of FID and CKA in the following table. Concretely, we use the full FFHQ dataset ($70K$ images) as the reference distribution and generate $50K$ images for evaluation, the clock time is tested on a single 3090 24G GPU. We could tell that CKA takes shorter time than FID when the same amount of samples are calculated.

| Extractor | Inception | ViT |
|-----------|-----------|-----------|
| FID | 3426 (s) | 3630 (s) |
| CKA | 3225 (s) | 3328 (s) |

**Q9: What is the theoretical sample complexity of CKA? Are there any known results here?**

**Reply**: Thanks. To the best of our knowledge, there are no known results of the theoretical sample complexity of CKA. CKA measures the distributional discrepancies between different distributions with a considerable samples from each distribution. Accordingly, involving sufficient samples for evaluation ensures more accurate results in practice. However, through evaluating the CKA scores under various data regimes, we observe that CKA shows satisfactory sample-efficiency and stability under different number of samples. Therefore, we can synthesize subsets with fixed number of images (*e.g.,* $50\,K$) for evaluation. By contrast, the FID and KID scores could be improved by producing more samples, which is unacceptable for a reliable evaluation.

**Q10: Is centered kernel alignment somehow related to the kernel maximum mean discrepancy (KID/MMD)?**

**Reply**: Thanks. Centered Kernel Alignment (CKA) is normalized from Hilbert-Schmidt Independence Criterion (HSIC) [5] to ensure it is invariant to isotropic scaling and is formally defined by

$$\text{CKA(X, Y)} = \frac{\text{HSIC}(x, y)}{\sqrt{\text{HSIC}(x, x)\text{HSIC}(y, y)}}. \tag{3}$$

HSIC is equivalent to maximum mean discrepancy (MMD) between the joint distribution and the product of the marginal distributions, and HSIC with a specific kernel family is equivalent to distance covariance [17]. HSIC determines whether two distributions are independent, and HSIC = 0 implies independence. However, HSIC is not invariant to isotropic scaling, making it sensitive to isotropic transformation of images when used for synthesis evaluation.

Hope that the above discussions could address your concerns, please let us know if you have any further questions. Thanks for your effort and constructive suggestions again.

**To Reviewer 95Dj**

**Q1: There are no ablation studies to separately prove the effectiveness of six extractors and CKA.**

**Reply**: Thanks. In this work, we seek to develop a new measurement system that could provide reliable and consistent synthesis comparisons. In particular, two key components are crucial for the measurement system, *i.e.,* the feature extractor that defines representation space and the distributional distance that deliver similarities. Accordingly, we make in-depth analyses on the reliability and robustness of various feature extractors and different distributional distances. For the feature extractors, we gather multiple models that are pre-trained with different objectives (fully-supervised/self-supervised) and various architectures (CNN/ViT/MLP). Notably, these models are chosen for a systematic investigation to comprehensively understand the intrinsic properties of various extractors, rather than based on existing findings. Then, we testify their performance on 1) how many semantic features they can capture for evaluation, 2) how robust they are when being attacked by the histogram matching mechanism, and 3) how distinct the representation space they can define. These investigations provide several new findings to the community, including 1) one specific extractor can only capture limited semantics and provide ons-side results, 2) extractors that are vulnerable

Table 4: **Quantitative comparison results of Fréchet Distance (FD$_\downarrow$) on ImageNet dataset**. "Random, Chosen$_I$" respectively represent the synthesized distribution of randomly generated and matching the class prediction of Inception-V3. Moreover, "$_v$" and "$_v$" respectively denote the architecture of ResNet and ViT. ($\downarrow$) indicates the results are hacked by the histogram matching mechanism. Notably, the values across different rows are not comparable and the results are tested three times.

| Model | Inception | ConvNeXt | SWAV | MoCo$_r$ | RepVGG | CLIP$_r$ | Swin | ViT | DeiT | CLIP$_v$ | MoCo$_v$ | ResMLP |
|---|---|---|---|---|---|---|---|---|---|---|---|---|
| Random | $34.29_{\pm0.09}$ | $78.02_{\pm0.16}$ | $0.13_{\pm0.003}$ | $0.32_{\pm0.002}$ | $54.98_{\pm0.22}$ | $27.64_{\pm0.15}$ | $323.12_{\pm0.88}$ | $50.97_{\pm0.20}$ | $621.98_{\pm1.02}$ | $5.46_{\pm0.09}$ | $50.01_{\pm0.21}$ | $145.32_{\pm1.02}$ |
| Chosen$_I$ | $33.05_{\pm0.08\downarrow}$ | $78.10_{\pm0.14}$ | $0.13_{\pm0.002}$ | $0.32_{\pm0.002}$ | $54.30_{\pm0.24}$ | $27.66_{\pm0.17}$ | $301.91_{\pm0.92\downarrow}$ | $50.96_{\pm0.18}$ | $597.32_{\pm1.11\downarrow}$ | $5.46_{\pm0.07}$ | $50.00_{\pm0.19}$ | $133.06_{\pm1.09\downarrow}$ |

to the histogram matching attack are not reliable, and 3) different feature extractors might define similar representation spaces. For the distributional distances, we investigate the numerical stability of different distances across various representation spaces and the sample efficiency of different distances. Through extensive comparisons, we find that Centered Kernel Alignment (CKA) provides a better comparison across various extractors and hierarchical layers with its bounded score. Moreover, CKA is more sample-efficiency and exhibits better agreement with human visual judgment. Together with these findings, we build a new measurement system that can accurately reflect the synthesis performance. Following this line, the effectiveness of each feature extractors and CKA is identified in our main experiments. In particular, Fig. 1 of the main paper indicates that the chosen six feature extractors can incorporate more visual semantics for evaluation in a complementary manner. And Tab. 2 of the main paper demonstrates that each of the chosen extractors is robust towards the histogram attack. Furthermore, in Tab.4 of the main paper and Tab.4, 5, 6, 7, 8 of the supplementary material, we provide qualitative and quantitative results of each extractor from various semantic levels. These results also demonstrate the reliability of each extractor when used for synthesis evaluation. In addition to evaluating the robustness of these extractors on the FFHQ dataset, we further perform the same experiment on the ImageNet dataset. Tab. 4 presents the quantitative results. We can tell from these results that the chosen feature extractors are robust to the attack, further demonstrating their reliability.

**Q2: It is important to research how to improve the speed of evaluation without affecting the evaluation accuracy.**

**Reply**: Thanks. We agree. Both evaluation speed and accuracy are very important in practice. This work focuses on developing a measurement system that could reliably and consistently reflect the synthesis performance. Based on the findings that one certain feature extractor might capture only limited semantics for evaluation, we integrate multiple extractors to alleviate this. Therefore, the evaluation time is relatively longer than using one extractor for evaluation. However, the inference time of these feature extractors is much shorter than the inference time of diffusion models. For instance, the evaluation time of our measurement system on $50K$ images is about 5 hours on a single 3090 24G GPU, but it takes about several days to generate $50K$ images with diffusion models (about 4 days for MDT and 2.5 days for DG-Diffusion). Consequently, improving both the speed of our evaluation and the inference speed of diffusion models is also important. In the future, we plan to integrate various accelerate techniques to improve our evaluation speed without compromising the evaluation accuracy, such as optimizing the model architecture, model pruning and distillation, *etc.*

**Q3: The layers of Section 2.3, 3.1 and 3.2 are not prominent and the organization of them is not clear. Specifically, the summary sentences are not emphasized and paragraph are not strictly parallel.**

**Reply**: Thanks. Our presentation is organized for following reasons: In Section 2.3, we present the details of generative models, evaluated datasets, and analysis approaches (including our visualization tool, histogram matching attack, and human evaluation). They are independent of each other, thus we discuss them in parallel in the main paper. In Section 3.1, we investigate the feature extractors by first identifying their attention on visual semantics, followed by investigating their robustness to the histogram matching attack. Finally, we filter extractors that define similar representation spaces. These studies are gradually deepening, thus they are organized in a progressive manner. In Section 3.2, we first study the numerical scales of CKA and FID across various extractors and hierarchical layers of one certain extractor. After that, we investigate the sample efficiency of CKA and KID. In

the last paragraph of Section 3.2, we summarize our findings about the feature extractors and the distributional distances. Moreover, the summary sentences of each paragraph provide our primary findings of this paragraph. Following your valuable suggestions, we will carefully proofread and revise the corresponding presentation to make our paper more logical.

Hope that the above discussions could address your concerns, please let us know if you have any further questions. Thanks for your effort and constructive suggestions again.

**To Reviewer y8MJ**

**Q1: The novelty is limited. CKA is a well-known metric for evaluating the similarity between distributions.**

**Reply**: Thanks. In this work, we seek to develop a new measurement system that could provide reliable and consistent synthesis comparisons. In particular, two key components are crucial for the measurement system, *i.e.,* the feature extractor that defines representation space and the distributional distance that deliver similarities. Accordingly, we make in-depth analyses on the reliability and robustness of various feature extractors and different distributional distances. For the feature extractors, we gather multiple models that are pre-trained with different objectives (fully-supervised/self-supervised) and various architectures (CNN/ViT/MLP). Then, we testify their performance on 1) how many semantic features they can capture for evaluation, 2) how robust they are when being attacked by the histogram matching mechanism, and 3) how distinct the representation space they can define. These investigations provide several new findings to the community, including 1) one specific extractor can only capture limited semantics and provide ons-side results, 2) extractors that are vulnerable to the histogram matching attack are not reliable, and 3) different feature extractors might define similar representation spaces. For the distributional distances, we investigate the numerical stability of different distances across various representation spaces and the sample efficiency of different distances. Through extensive comparisons, we find that Centered Kernel Alignment (CKA) provides a better comparison across various extractors and hierarchical layers with its bounded score. Moreover, CKA is more sample-efficiency and exhibits better agreement with human visual judgment. Together with these findings, we build a new measurement system that can accurately reflect the synthesis performance. To the best of our knowledge, this paper is the first work to present these findings about feature extractors and to incorporate CKA for synthesis measurement in the community. We believe that these findings can provide potential insights to further works that develop new evaluation protocols.

**Q2: Lacks discussion of some state-of-the-art methods, such as stable diffusion and midjourney.**

**Reply**: Thanks. Following your valuable suggestion, we further include more state-of-the-art generative models on the ImageNet dataset for synthesis evaluation, namely GigaGAN (CVPR'2023) [7], MDT (ICCV'2023) [4], and DG-Diffusion (ICML'2023) [8]. Specifically, we either gather their official models for inference or download the pre-generated images released by the authors for evaluation. Similarly, $50K$ generated images and the entire training set (*i.e.,* $1.28M$ images) are used as the synthesized and real distributions, respectively. All details are consistent with the experiments conducted in our main paper. Tab. 5 presents the quantitative comparison results. Akin to the results in our main paper, our evaluation system provides consistent ranks with FID and human visual evaluation, demonstrating the reliability of our metric. Notably, this paper focuses on evaluating the performance of various generative models trained on single modality (*i.e.,* images). Therefore, evaluating generative models that are trained on multiple modality synthesis tasks (*e.g.,* text-to-image generation) is slightly out of our scope. However, multiple modality tasks such as text-to-image/video have made remarkable progress recently, and evaluating their performance accurately is a very important and promising topic. Accordingly, we plan to investigate the performance of our measurement system under multiple modality synthesis tasks in our future work.

**Q3: This paper only compares CKA with FID and lacks a comparison with the other metrics. A discussion of these related metrics is needed.**

**Reply**: Thanks. Following your valuable suggestion, we further involve Kernel Inception Distance (KID) [1], precision, and recall [15] into our comparison. Note that the original KID employs Inception-V3 as the feature extractor, and there is a large "perceptual null space" in Inception-V3.

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

**Q4: It will be good if the reviewer can see the dataset during the review process.**

**Reply**: Thanks. All evaluated datasets and generative models are publicly available thanks to the original authors' generous release. For synthesized images, we either gather the pre-computed datasets from the official repositories or use public models with the official settings to generate new

images for evaluation. We will make our code and evaluation scripts publicly available, making it easier to evaluate synthesis performance.

Hope that the above discussions could address your concerns, please let us know if you have any further questions. Thanks for your effort and constructive suggestions again.

**To Reviewer xEcW**

**Q1: It would be beneficial to have a proposed metric for evaluating the performance of image translation.**

**Reply**: Thanks. Following your valuable suggestion, we employ our measurement system to evaluate the performance of image-to-image translation. We collect publicly available image-to-image translation models that are officially released to translate images from one domain to another domain for evaluation. Specifically, three translation benchmarks are involved here, namely Horse-to-Zebra [19, 23, 13], Cat-to-Dog [21, 13], and Dog-to-Cat [21, 6]. For each benchmark, we translate the tested images to the target domain following the original experimental settings. Then we compute the distributional discrepancies between the translated images and the real target images. Tab. 7, 8, and Tab. 9 respectively present the quantitative results of the evaluated three image-to-image translation benchmarks. It can be seen from these results that CKA provides consistent ranks with FID among various extractors, and the averaged score can reflect the performance of different image translation models. For instance, the performance of CUT [13] on Horse-to-Zebra is identified better than that of CycleGAN [23] by both FID and our proposed metric. And the qualitative results in the original paper of CUT [13] also suggest that the performance of CUT surpasses CycleGAN. That is, our measurement system can provide a reliable evaluation under such settings. This indicates that our measurement system can also be used for evaluating the performance of image translation tasks. These results will be added in the next version of our paper, and we plan involve more state-of-the-art image translation models for evaluation for future work.

Table 7: **Quantitative comparison results of Centered Kernel Alignment (CKA$_\uparrow$) on Horse-to-Zebra dataset**.

| Model | FID | ConvNeXt | RepVGG | SWAV | ViT | MoCo-ViT | CLIP-ViT | Overall |
|---|---|---|---|---|---|---|---|---|
| CycleGAN [23] | 83.32 | 73.55 | 88.67 | 85.82 | 83.96 | 74.72 | 73.74 | 80.08 |
| AttentionGAN [19] | 76.05 | 75.59 | 91.73 | 86.37 | 85.16 | 76.65 | 75.49 | 81.83 |
| CUT [13] | 51.29 | 78.48 | 93.22 | 88.83 | 87.84 | 78.75 | 77.36 | 84.08 |

Table 8: **Quantitative comparison results of Centered Kernel Alignment (CKA$_\uparrow$) on Cat-to-Dog dataset**.

| Model | FID | ConvNeXt | RepVGG | SWAV | ViT | MoCo-ViT | CLIP-ViT | Overall |
|---|---|---|---|---|---|---|---|---|
| CUT [13] | 74.95 | 84.93 | 78.75 | 88.83 | 84.31 | 93.56 | 70.91 | 83.55 |
| GP-UNIT [21] | 60.96 | 90.45 | 87.79 | 94.05 | 90.12 | 95.91 | 75.32 | 88.94 |

Table 9: **Quantitative comparison results of Centered Kernel Alignment (CKA$_\uparrow$) on Dog-to-Cat dataset**.

| Model | FID | ConvNeXt | RepVGG | SWAV | ViT | MoCo-ViT | CLIP-ViT | Overall |
|---|---|---|---|---|---|---|---|---|
| GP-UNIT [21] | 31.66 | 79.58 | 78.18 | 96.79 | 86.93 | 93.92 | 77.42 | 85.47 |
| MUNIT [6] | 18.88 | 84.87 | 84.11 | 98.51 | 88.11 | 95.95 | 86.10 | 89.61 |

**Q2: Although a large amount of experiments has been conducted, this work seems to simply exploited existing findings such as advantages of CNN-transformer networks.**

**Reply**: Thanks. In this work, we seek to develop a new measurement system that could provide reliable and consistent synthesis comparisons. In particular, two key components are crucial for the measurement system, *i.e.,* the feature extractor that defines representation space and the distributional distance that deliver similarities. Accordingly, we make in-depth analyses on the reliability and robustness of various feature extractors and different distributional distances. For the feature extractors, we gather multiple models that are pre-trained with different objectives (fully-supervised/self-supervised) and various architectures (CNN/ViT/MLP). Notably, these models are

chosen for a systematic investigation to comprehensively understand the intrinsic properties of various extractors, rather than based on existing findings. Then, we testify their performance on 1) how many semantic features they can capture for evaluation, 2) how robust they are when being attacked by the histogram matching mechanism, and 3) how distinct the representation space they can define. These investigations provide several new findings to the community, including 1) one specific extractor can only capture limited semantics and provide ons-side results, 2) extractors that are vulnerable to the histogram matching attack are not reliable, and 3) different feature extractors might define similar representation spaces. For the distributional distances, we investigate the numerical stability of different distances across various representation spaces and the sample efficiency of different distances. Through extensive comparisons, we find that Centered Kernel Alignment (CKA) provides a better comparison across various extractors and hierarchical layers with its bounded score. Moreover, CKA is more sample-efficiency and exhibits better agreement with human visual judgment. Together with these findings, we build a new measurement system that can accurately reflect the synthesis performance. To the best of our knowledge, this paper is the first work to present these findings in the community of generative models. We believe that these findings can provide potential insights to further works that develop new evaluation protocols.

**Q3: In addition to quality, this study should extend the metric to assess the diversity and novelty of generated samples.**

**Reply**: Thanks. The target of generative models is to reproduce the observed data distribution, thus a good metric should accurately deliver the distributional discrepancy between the synthesized distribution and the real distribution to reflect the synthesis performance. Accordingly, our proposed evaluation system focuses on capturing the similarity between different data distributions instead of one certain aspect of the synthesized images, *e.g.,* quality and fidelity. By comparing the distributional distances between the original distribution and the synthesized distribution produced by various generative models, we can capture their actual improvement . We agree that assessing the diversity and novelty of generated samples is crucial to understand the intrinsic properties of the synthesized distributions, but this is slightly out of scope in this paper. We plan to investigate the performance of our evaluation system in assessing the synthesis diversity and novelty in our future studies as suggested.

Hope that the above discussions could address your concerns, please let us know if you have any further questions. Thanks for your effort and constructive suggestions again.

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

# Revisiting the Evaluation of Image Synthesis with GANs — *Supplementary Material*

**Anonymous Author(s)**
Affiliation
Address
`email`

This *Supplementary Material* is organized as follows: appendix A discusses the limitations of our paper and appendix B provides the implementation details of our experiments, appendix C demonstrates how human visual judgment is performed, and appendix D presents more quantitative and qualitative results.

## A    Limitations

Despite a comprehensive investigation, our study could still be extended in several aspects. For instance, the impacts of different low-level image processing techniques (*e.g.,* resizing) could be identified since they also play an important role in synthesis evaluation [11]. Besides, comparing datasets with various resolutions could be further studied. Nonetheless, our study could be considered an empirical revisiting towards the paradigm of evaluating generative models. We hope this work could inspire more fascinating works of synthesis evaluation and provide potential insight to develop more comprehensive evaluation protocols. We will also conduct more investigation on the unexplored factors and compare more generative models with our system.

## B    Implementation Details

### B.1    Datasets

**FFHQ** [14] contains unique $70,000$ human-face images with large variations in terms of age, ethnicity, and facial expressions. We employ the resolution of $256 \times 256 \times 3$ for our experiments.

**ImageNet** [4] includes $1,280,000$ images with $1,000$ classes of different objects such as goldfish, bow tie, etc. All experiments on ImageNet are performed with the resolution of $256 \times 256 \times 3$ unless otherwise specified.

**LSUN Church** [17] consists of $126,227$ images of the church, varies in the background, perspectives, etc. We employ the resolution of $256 \times 256 \times 3$ for our experiments.

### B.2    Experimental Settings and Hyperparameters

**Kernel selection**. We consistently employ the RBF kernel

$$K(\mathbf{x_i}, \mathbf{x_j}) = \exp(-\frac{\|\mathbf{x_i} - \mathbf{x_j}\|^2}{2\sigma^2})$$

for calculating the CKA. The bandwidth $\sigma$ is set as a fraction of the median distance between examples. In practice, three commonly used kernels could be employed for calculation, namely linear, polynomial, and RBF kernels. In order to investigate their difference, three publicly available

Submitted to the 37th Conference on Neural Information Processing Systems (NeurIPS 2023) Track on Datasets and Benchmarks. Do not distribute.