# OpenReview forum: "Revisiting the Evaluation of Image Synthesis with GANs"
_NeurIPS.cc/2023/Track/Datasets_and_Benchmarks — NeurIPS 2023 Datasets and Benchmarks Poster_

### Official Review · Reviewer_xEcW · 2023-07-13
**Lack of Original Contribution in Dataset and Benchmark Design, raising doubts about the paper's heavily relies on existing methodologies**

**Rating:** 5
**Confidence:** 4

**Strengths:**

- The FID metric, which evaluates the quality of generative models, has been used for a long time. The paper aims to address the problem of feature bias that emerges from relying on a single extractor. In order to ensure reliable measurement, the paper utilize various extractors to capture different aspects of a Semantic features. Furthermore,  the paper provides empirical evidence supporting the high reliability of the Centered Kernel Alignment (CKA) as a distance measure in diverse scenarios.

**Additional Feedback:**

- N/A

**Clarity:**

- The paper is well-organized.

**Correctness:**

- Yes, experiments look designed and performed correctly.

**Documentation:**

- There is sufficient detail to support reproducibility.

**Ethics:**

- N/A

**Limitations:**

- **Although a large amount of experiments has been conducted, this work seems to simply exploited existing findings such as advantages of CNN-transformer networks**
- **There are drawbacks to solely relying on fidelity in evaluating generative models. In** **addition to quality, this study should extend the metric to assess the diversity and novelty of generated samples.**

**Opportunities For Improvement:**

- It would be beneficial to have a proposed metric for evaluating the performance of image translation. For instance, translation performance can be measured by introducing the  relative distance which indicate how far from the source domain and how close to the target domain.

**Relation To Prior Work:**

- Yes. The paper discusses previous benchmarks and compares them well.

**Summary And Contributions:**

**The paper presents an empirical investigation to evaluate the synthesis performance with generative adversarial networks (GANs).**

**The key contributions are:**

- **Hybrid architectures (CNN+ViT) serve as reliable and robust feature extractors**
- **Centered Kernel Alignment (CKA) provides a better comparison across various extractors and hierarchical layers in one model**
- **Extensive experiments conducted on multiple datasets**

---

> ### Author Response · Authors · 2023-08-21
> **Response to Reviewer xEcW (Part1)**
>
> Thanks for the valuable comments. Individual concerns are addressed as follows. Note that, to make sure the revision is easy to track, we do not change the paper structure, and instead list all additional results in the rebuttal of the supplementary material. We will rearrange these materials in the next version. We appreciate the constructive feedback, which has helped to improve the clarity and rigor of our work.
>
> **Q1. It would be beneficial to have a proposed metric for evaluating the performance of image translation.**
>
> **Reply:**
> Thanks.
> Following your valuable suggestion, we employ our measurement system to evaluate the performance of image-to-image translation.
> We collect publicly available image-to-image translation models that are officially released to translate images from one domain to another domain for evaluation.
> Specifically, three translation benchmarks are involved here, namely Horse-to-Zebra [1] [2] [3] , Cat-to-Dog [3] [4], and Dog-to-Cat [4] [5].
> For each benchmark, we translate the tested images to the target domain following the original experimental settings.
> Then we compute the distributional discrepancies between the translated images and the real target images.
> The following tables respectively present the quantitative results of the evaluated three image-to-image translation benchmarks.
> It can be seen from these results that CKA provides consistent ranks with FID among various extractors, and the averaged score can reflect the performance of different image translation models.
> For instance, the performance of CUT on Horse-to-Zebra is identified better than that of CycleGAN by both FID and our proposed metric.
> And the qualitative results in the original paper of CUT also suggest that the performance of CUT surpasses CycleGAN.
> That is, our measurement system can provide a reliable evaluation under such settings.
> This indicates that our measurement system can also be used for evaluating the performance of image translation tasks.
> These results will be added in the next version of our paper, and we plan involve more state-of-the-art image translation models for evaluation for future work.
>
> **Horse-to-Zebra:**
> | Model                                    | FID   | ConvNeXt | RepVGG | SWAV  | ViT   | MoCo-ViT | CLIP-ViT | Overall |
> |------------------------------------------|-------|----------|--------|-------|-------|----------|----------|---------|
> | CycleGAN [1]          | 83.32 | 73.55    | 88.67  | 85.82 | 83.96 | 74.72    | 73.74    | 80.08   |
> | AttentionGAN [2] | 76.05 | 75.59    | 91.73  | 86.37 | 85.16 | 76.65    | 75.49    | 81.83   |
> | CUT [3]          | 51.29 | 78.48    | 93.22  | 88.83 | 87.84 | 78.75    | 77.36    | 84.08   |
>
> **Cat-to-Dog:**
> | Model                               | FID   | ConvNeXt | RepVGG | SWAV  | ViT   | MoCo-ViT | CLIP-ViT | Overall |
> |-------------------------------------|-------|----------|--------|-------|-------|----------|----------|---------|
> | CUT [4]     | 74.95 | 84.93    | 78.75  | 88.83 | 84.31 | 93.56    | 70.91    | 83.55   |
> | GP-UNIT [4] | 60.96 | 90.45    | 87.79  | 94.05 | 90.12 | 95.91    | 75.32    | 88.94   |
>
> **Dog-to-Cat:**
> | Model                               | FID   | ConvNeXt | RepVGG | SWAV  | ViT   | MoCo-ViT | CLIP-ViT | Overall |
> |-------------------------------------|-------|----------|--------|-------|-------|----------|----------|---------|
> | GP-UNIT [4] | 31.66 | 79.58    | 78.18  | 96.79 | 86.93 | 93.92    | 77.42    | 85.47   |
> | MUNIT [5]   | 18.88 | 84.87    | 84.11  | 98.51 | 88.11 | 95.95    | 86.10    | 89.61   |
>
> [1]  Zhu, Jun-Yan and Park, Taesung and Isola, Phillip and Efros, Alexei A. 2017. Unpaired image-to-image translation using cycle-consistent adversarial networks. *ICCV*.
>
> [2] Tang, Hao and Liu, Hong and Xu, Dan and Torr, Philip HS and Sebe, Nicu. 2021. Attentiongan: Unpaired image-to-image translation using attention-guided generative adversarial networks. *TNNLS*.
>
> [3] Park, Taesung and Efros, Alexei A and Zhang, Richard and Zhu, Jun-Yan. 2020. Contrastive learning for unpaired image-to-image translation. *ECCV*.
>
> [4] Yang, Shuai and Jiang, Liming and Liu, Ziwei and Loy, Chen Change. 2022. Unsupervised image-to-image translation with generative prior. *CVPR*.
>
> [5] Huang, Xun and Liu, Ming-Yu and Belongie, Serge and Kautz, Jan. 2018. Multimodal unsupervised image-to-image translation. *ECCV*.

---

> > ### Author Response · Authors · 2023-08-21
> > **Response to Reviewer xEcW (Part2)**
> >
> > **Q2. Although a large amount of experiments has been conducted, this work seems to simply exploited existing findings such as advantages of CNN-transformer networks.**
> >
> > **Reply:**
> > Thanks.
> > In this work, we seek to develop a new measurement system that could provide reliable and consistent synthesis comparisons.
> > In particular, two key components are crucial for the measurement system, *i.e.*, the feature extractor that defines representation space and the distributional distance that deliver similarities.
> > Accordingly, we make in-depth analyses on the reliability and robustness of various feature extractors and different distributional distances.
> > For the feature extractors, we gather multiple models that are pre-trained with different objectives (fully-supervised/self-supervised) and various architectures (CNN/ViT/MLP).
> > Notably, these models are chosen for a systematic investigation to comprehensively understand the intrinsic properties of various extractors, rather than based on existing findings.
> > Then, we testify their performance on 1) how many semantic features they can capture for evaluation, 2) how robust they are when being attacked by the histogram matching mechanism, and 3) how distinct the representation space they can define.
> > These investigations provide several new findings to the community, including 1) one specific extractor can only capture limited semantics and provide ons-side results, 2) extractors that are vulnerable to the histogram matching attack are not reliable, and 3) different feature extractors might define similar representation spaces.
> > For the distributional distances, we investigate the numerical stability of different distances across various representation spaces and the sample efficiency of different distances.
> > Through extensive comparisons, we find that Centered Kernel Alignment (CKA) provides a better comparison across various extractors and hierarchical layers with its bounded score.
> > Moreover, CKA is more sample-efficiency and exhibits better agreement with human visual judgment.
> >
> > Together with these findings, we build a new measurement system that can accurately reflect the synthesis performance.
> > To the best of our knowledge, this paper is the first work to present these findings in the community of generative models.
> > We believe that these findings can provide potential insights to further works that develop new evaluation protocols.
> >
> > **Q3. In addition to quality, this study should extend the metric to assess the diversity and novelty of generated samples.**
> >
> > **Reply:**
> > Thanks.
> > The target of generative models is to reproduce the observed data distribution, thus a good metric should accurately deliver the distributional discrepancy between the synthesized distribution and the real distribution to reflect the synthesis performance.
> > Accordingly, our proposed evaluation system focuses on capturing the similarity between different data distributions instead of one certain aspect of the synthesized images, *e.g.*, quality and fidelity.
> > By comparing the distributional distances between the original distribution and the synthesized distribution produced by various generative models, we can capture their actual improvement .
> > We agree that assessing the diversity and novelty of generated samples is crucial to understand the intrinsic properties of the synthesized distributions, but this is slightly out of scope in this paper.
> > We plan to investigate the performance of our evaluation system in assessing the synthesis diversity and novelty in our future studies as suggested.
> >
> > Hope that the above discussions could address your concerns, please let us know if you have any further questions. Thanks for your effort and constructive suggestions again.

---

> ### Author Response · Authors · 2023-08-26
> **Response to Reviewer xEcW**
>
> Dear Reviewer xEcW,
>
> Thank you so much for your effort in reviewing this submission! We have tried our best to address the mentioned concerns/problems in the rebuttal. Please let us know if you have any additional questions, we are happy to clarify them, thanks for your time and effort again.
>
> Best, Authors.

---

### Official Review · Reviewer_y8MJ · 2023-07-20
**Review of Paper340**

**Rating:** 6
**Confidence:** 4
**Correctness:** Yes
**Clarity:** See Opportunities For Improvement

**Strengths:**

1. this paper conducts many experiments to show the effectiveness of CKA in the synthesis task.
2. The paper is clearly written, well organized, and easy to follow.


**Additional Feedback:**

N/A

**Documentation:**

Lacks a URL for reviewer access to the dataset. The authors promise that the dataset will be publicly available later this year. It will be good if the reviewer can see the dataset during the review process.

**Ethics:**

No concerns

**Limitations:**

The authors have addressed the limitations.

**Opportunities For Improvement:**

1. The novelty is limited. Cka is a well-known metric for evaluating the similarity between distributions. This paper only evaluates CKA in the synthesis task, without any novelty design related to the task.
2. Lacks discussion of some state-of-the-art methods, such as stable diffusion and midjourney.
3. this paper only compares CKA with FID and lacks a comparison with the other metrics. A discussion of these related metrics is needed.


**Relation To Prior Work:**

Yes

**Summary And Contributions:**

The paper proposes a new metric named CKA to evaluate the synthesis performance of generative models. This paper shows that a group of models are more reliable to be feature extractors compared with a single model. Empirical experiments are conducted to show CKA has better correlations with human perceptual evaluation than FID.

---

> ### Author Response · Authors · 2023-08-21
> **Response to Reviewer y8MJ (Part1)**
>
> Thanks for the valuable comments. Individual concerns are addressed as follows. Note that, to make sure the revision is easy to track, we do not change the paper structure, and instead list all additional results in the rebuttal of the supplementary material. We will rearrange these materials in the next version. We appreciate the constructive feedback, which has helped to improve the clarity and rigor of our work.
>
> **Q1. The novelty is limited. CKA is a well-known metric for evaluating the similarity between distributions.**
>
> **Reply:**
> Thanks.
> In this work, we seek to develop a new measurement system that could provide reliable and consistent synthesis comparisons.
> In particular, two key components are crucial for the measurement system, *i.e.*, the feature extractor that defines representation space and the distributional distance that deliver similarities.
> Accordingly, we make in-depth analyses on the reliability and robustness of various feature extractors and different distributional distances.
> For the feature extractors, we gather multiple models that are pre-trained with different objectives (fully-supervised/self-supervised) and various architectures (CNN/ViT/MLP).
> Then, we testify their performance on 1) how many semantic features they can capture for evaluation, 2) how robust they are when being attacked by the histogram matching mechanism, and 3) how distinct the representation space they can define.
> These investigations provide several new findings to the community, including 1) one specific extractor can only capture limited semantics and provide ons-side results, 2) extractors that are vulnerable to the histogram matching attack are not reliable, and 3) different feature extractors might define similar representation spaces.
> For the distributional distances, we investigate the numerical stability of different distances across various representation spaces and the sample efficiency of different distances.
> Through extensive comparisons, we find that Centered Kernel Alignment (CKA) provides a better comparison across various extractors and hierarchical layers with its bounded score.
> Moreover, CKA is more sample-efficiency and exhibits better agreement with human visual judgment.
>
> Together with these findings, we build a new measurement system that can accurately reflect the synthesis performance.
> To the best of our knowledge, this paper is the first work to present these findings about feature extractors and to incorporate CKA for synthesis measurement in the community.
> We believe that these findings can provide potential insights to further works that develop new evaluation protocols.
>
> **Q2. Lacks discussion of some state-of-the-art methods, such as stable diffusion and midjourney.**
>
> **Reply:**
> Thanks.
> Following your valuable suggestion, we further include more state-of-the-art generative models on the ImageNet dataset for synthesis evaluation, namely GigaGAN (CVPR'2023), MDT (ICCV'2023), and DG-Diffusion (ICML'2023).
> Specifically, we either gather their official models for inference or download the pre-generated images released by the authors for evaluation.
> Similarly, 50$K$ generated images and the entire training set (*i.e.*, 1.28$M$ images) are used as the synthesized and real distributions, respectively.
> All details are consistent with the experiments conducted in our main paper.
> The following table presents the quantitative comparison results.
> Akin to the results in our main paper, our evaluation system provides consistent ranks with FID and human visual evaluation, demonstrating the reliability of our metric.
> Notably, this paper focuses on evaluating the performance of various generative models trained on single modality (\emph{i.e.,} images).
> Therefore, evaluating generative models that are trained on multiple modality synthesis tasks (*e.g.*, text-to-image generation) is slightly out of our scope.
> However, multiple modality tasks such as text-to-image/video have made remarkable progress recently, and evaluating their performance accurately is a very important and promising topic.
> Accordingly, we plan to investigate the performance of our measurement system under multiple modality synthesis tasks in our future work.
>
> | Model                               | FID$^\dagger$ | ConvNeXt | RepVGG | SWAV  | ViT   | MoCo-ViT | CLIP-ViT | Overall | User study |
> |-------------------------------------|---------------|----------|--------|-------|-------|----------|----------|---------|------------|
> | GigaGAN             | 3.45          | 68.01    | 79.93  | 90.15 | 98.34 | 82.40    | 96.52    | 85.89   | 65\%       |
> | DG-Diffusion | 3.18          | 68.22    | 80.06  | 90.56 | 98.46 | 82.51    | 96.88    | 86.12   | 66\%       |
> | MDT           | 1.79          | 69.64    | 81.68  | 91.78 | 99.43 | 83.43    | 98.19    | 87.36   | 69\%       |

---

> > ### Author Response · Authors · 2023-08-21
> > **Response to Reviewer y8MJ (Part2)**
> >
> > **Q3. This paper lacks a comparison with the other metrics.**
> >
> > **Reply:**
> > Thanks.
> > Following your valuable suggestion, we further involve Kernel Inception Distance (KID), precision, and recall into our comparison.
> > Note that the original KID employs Inception-V3 as the feature extractor, and there is a large ``perceptual null space" in Inception-V3.
> > Therefore, we first investigate whether KID scores can be altered by attacking the feature extractor with the histogram matching mechanism.
> > The experimental details are consistent with computing Frechet Distance in Tab.2 of the main paper.
> > The following table presents the quantitative results.
> > Still, some extractors, such as Inception, Swin-Transformer, and ResMLP, are susceptible to the histogram matching attack.
> > For instance, the KID score of Swin-Transformer is improved by 5.31\% when the chosen set is used.
> > These observations agree with our findings in our main paper, suggesting that certain extractors can be hacked when KID is employed as the distributional distance.
> > Then, we investigate the sample efficiency of KID, Precision, and Recall to probe the impacts of the amount of generated samples.
> > Fig.5 in the rebuttal of the *Supplementary Material*  presents the curves of KID, Precision, and Recall scores computed under different data regimes.
> > Similarly, we could observe that the KID scores can be improved by synthesizing more images.
> > Interestingly, the recall scores decrease as the generated sample size increases whereas the precision is stable.
> > This is caused by the definition of recall: recall measures the proportion of the real distribution that is covered by the synthesized distribution.
> > In practical computation, the denominator increases as the synthesized samples increases, while the numerator (*i.e.*, images from the real distribution) remain unchanged.
> > In this way, the recall scores decrease as the generated sample size increases and vice versa.
> > By contrast, CKA scores are stable under different data regimes (please see Fig. 2 in the main paper).
> > Moreover, CKA can provide reliable synthesis evaluation that agrees with human visual judgment.
> > Accordingly, CKA is a proper choice for building a consistent and reliable measurement system.
> > These results will be added in the next version of our paper.
> >
> > | Model      | Inception                                               | ConvNeXt          | SWAV             | MoCo$_r$         | RepVGG            | CLIP$_r$         | Swin                                                     | ViT               | DeiT              | CLIP$_v$          | MoCo$_v$          | ResMLP                                                  |
> > |------------|---------------------------------------------------------|-------------------|------------------|------------------|-------------------|------------------|----------------------------------------------------------|-------------------|-------------------|-------------------|-------------------|---------------------------------------------------------|
> > | Random     | 1.88$\pm$0.02                                        | 34.81$\pm$0.11 | 9.61$\pm$0.06 | 5.31$\pm$0.06 | 33.88$\pm$0.29 | 2.85$\pm$0.05 | 21.64$\pm$0.10                                        | 16.74$\pm$0.10 | 18.01$\pm$0.19 | 38.06$\pm$0.20 | 15.41$\pm$0.09 | 4.86$\pm$0.02                                        |
> > | Chosen$_I$ | 1.71$\pm$0.02 | 34.82$\pm$0.10 | 9.61$\pm$0.06 | 5.31$\pm$0.05 | 33.89$\pm$0.27 | 2.85$\pm$0.05 | 20.49$\pm$0.09 | 16.74$\pm$0.12 | 19.39$\pm$0.22 | 38.09$\pm$0.19 | 15.40$\pm$0.07 | 4.70$\pm$0.02 |
> >
> > **Q4. It will be good if the reviewer can see the dataset.**
> >
> > **Reply:**
> > Thanks.
> > All evaluated datasets and generative models are publicly available thanks to the original authors' generous release.
> > For synthesized images, we either gather the pre-computed datasets from the official repositories or use public models with the official settings to generate new images for evaluation. We will make our code and evaluation scripts publicly available, making it easier to evaluate synthesis performance.
> >
> > Hope that the above discussions could address your concerns, please let us know if you have any further questions. Thanks for your effort and constructive suggestions again.

---

> ### Author Response · Authors · 2023-08-26
> **Response to Reviewer y8MJ**
>
> Dear Reviewer y8MJ,
>
> Thank you so much for your effort in reviewing this submission! We have tried our best to address the mentioned concerns/problems in the rebuttal. Please let us know if you have any additional questions, we are happy to clarify them, thanks for your time and effort again.
>
> Best, Authors.

---

### Official Review · Reviewer_95Dj · 2023-07-21
**Review of a new synthesis valuation system of image synthesis with GANs**

**Rating:** 7
**Confidence:** 3

**Strengths:**

1. The contribution of the paper is significant because of the proposed novel and effective method to evaluate the quality of synthesized images.

2. The paper involves image synthesis, feature extraction and distribution distance, which is relevant to a wide research community.

3. The ethical and social impact of the paper is positive as it can promote the development and application of image synthesis technology, creating a better and more diverse visual experience for humanity.

**Additional Feedback:**

No. Please refer to the review comments above.

**Clarity:**

Not very clear.

The layers of Section 2.3, 3.1 and 3.2 are not prominent and the organization of them is not clear. Specifically, the summary sentences are not emphasized and paragraph are not strictly parallel.

**Correctness:**

Not very appropriate.

When testing the new synthesis valuation system, there are no ablation studies to separately prove the effectiveness of six extractors and CKA.

**Documentation:**

Yes, the datasets and models used in experiments are all open source.

**Ethics:**

No.

**Limitations:**

The paper utilizes six different feature extractors to evaluate synthesized images, which may significantly reduce the speed of evaluation. In the future, it is important to research how to improve the speed of evaluation without affecting the evaluation accuracy.

**Opportunities For Improvement:**

The experiments are not adequate. Specifically, there is no ablation study to support the last conclusion, which greatly weakens persuasiveness.

**Relation To Prior Work:**

Yes. This paper proposes a new synthesis valuation system to evaluate generative models.

**Summary And Contributions:**

The paper proposes a new synthesis valuation system, which consists of feature extractors and distributional distance.
The system is more reliable and comprehensive for synthesis evaluation.
In this paper, reliable feature extractors are considered to capture rich semantics in a complementary way, be robust toward the histogram matching attack and define meaningful and distinctive representation spaces.
Based on that belief, six extractors, ConvNeXt, SWAV, RepVGG, CLIP-ViT, MoCo-ViT and ViT, are selected as reliable tools for synthesis evaluation.
This paper also discovers that CKA shows remarkable stability across various representation spaces and various layers of the neural networks.
Besides, CKA shows satisfactory sample-efficiency and stability under different number of samples.
The experiment results show that the new systhesis valuation system provides the right rankings and better correlations with human visual judgment than FID.

---

> ### Author Response · Authors · 2023-08-21
> **Response to Reviewer 95Dj (Part1)**
>
> Thanks for the valuable comments. Individual concerns are addressed as follows. Note that, to make sure the revision is easy to track, we do not change the paper structure, and instead list all additional results in the rebuttal of the supplementary material. We will rearrange these materials in the next version. We appreciate the constructive feedback, which has helped to improve the clarity and rigor of our work.
>
> **Q1. There are no ablation studies to separately prove the effectiveness of six extractors and CKA.**
>
> **Reply:** Thanks.
> In this work, we seek to develop a new measurement system that could provide reliable and consistent synthesis comparisons.
> In particular, two key components are crucial for the measurement system, *i.e.*, the feature extractor that defines representation space and the distributional distance that deliver similarities.
> Accordingly, we make in-depth analyses on the reliability and robustness of various feature extractors and different distributional distances.
> For the feature extractors, we gather multiple models that are pre-trained with different objectives (fully-supervised/self-supervised) and various architectures (CNN/ViT/MLP).
> Notably, these models are chosen for a systematic investigation to comprehensively understand the intrinsic properties of various extractors, rather than based on existing findings.
> Then, we testify their performance on 1) how many semantic features they can capture for evaluation, 2) how robust they are when being attacked by the histogram matching mechanism, and 3) how distinct the representation space they can define.
> These investigations provide several new findings to the community, including 1) one specific extractor can only capture limited semantics and provide ons-side results, 2) extractors that are vulnerable to the histogram matching attack are not reliable, and 3) different feature extractors might define similar representation spaces.
> For the distributional distances, we investigate the numerical stability of different distances across various representation spaces and the sample efficiency of different distances.
> Through extensive comparisons, we find that Centered Kernel Alignment (CKA) provides a better comparison across various extractors and hierarchical layers with its bounded score.
> Moreover, CKA is more sample-efficiency and exhibits better agreement with human visual judgment.
>
> Together with these findings, we build a new measurement system that can accurately reflect the synthesis performance.
> Following this line, the effectiveness of each feature extractors and CKA is identified in our main experiments.
> In particular, Fig. 1 of the main paper indicates that the chosen six feature extractors can incorporate more visual semantics for evaluation in a complementary manner.
> And Tab. 2 of the main paper demonstrates that each of the chosen extractors is robust towards the histogram attack.
> Furthermore, in Tab.4 of the main paper and Tab.4, 5, 6, 7, 8 of the supplementary material, we provide qualitative and quantitative results of each extractor from various semantic levels.
> These results also demonstrate the reliability of each extractor when used for synthesis evaluation.
> In addition to evaluating the robustness of these extractors on the FFHQ dataset, we further perform the same experiment on the ImageNet dataset.
> The following table presents the quantitative results.
> We can tell from these results that the chosen feature extractors are robust to the attack, further demonstrating their reliability.
>
> | Model      | Inception                                                | ConvNeXt          | SWAV              | MoCo$_r$          | RepVGG            | CLIP$_r$          | Swin                                                      | ViT               | DeiT                                                      | CLIP$_v$         | MoCo$_v$          | ResMLP                                                    |
> |------------|----------------------------------------------------------|-------------------|-------------------|-------------------|-------------------|-------------------|-----------------------------------------------------------|-------------------|-----------------------------------------------------------|------------------|-------------------|-----------------------------------------------------------|
> | Random     | 34.29$\pm$0.09                                        | 78.02$\pm$0.16 | 0.13$\pm$0.003 | 0.32$\pm$0.002 | 54.98$\pm$0.22 | 27.64$\pm$0.15 | 323.12$\pm$0.88                                        | 50.97$\pm$0.20 | 621.98$\pm$1.02                                        | 5.46$\pm$0.09 | 50.01$\pm$0.21 | 145.32$\pm$1.02                                        |
> | Chosen$_I$ | 33.05$\pm$0.08 | 78.10$\pm$0.14 | 0.13$\pm$0.002 | 0.32$\pm$0.002 | 54.30$\pm$0.24 | 27.66$\pm$0.17 | 301.91$\pm$0.92 | 50.96$\pm$0.18 | 597.32$\pm$1.11 | 5.46$\pm$0.07 | 50.00$\pm$0.19 | 133.06$\pm$1.09 |

---

> > ### Author Response · Authors · 2023-08-21
> > **Response to Reviewer 95Dj (Part2)**
> >
> > **Q2. It is important to research how to improve the speed of evaluation without affecting the evaluation accuracy.**
> >
> > **Reply:**
> > Thanks.
> > We agree.
> > Both evaluation speed and accuracy are very important in practice.
> > This work focuses on developing a measurement system that could reliably and consistently reflect the synthesis performance.
> > Based on the findings that one certain feature extractor might capture only limited semantics for evaluation, we integrate multiple extractors to alleviate this.
> > Therefore, the evaluation time is relatively longer than using one extractor for evaluation.
> > However, the inference time of these feature extractors is much shorter than the inference time of diffusion models.
> > For instance, the evaluation time of our measurement system on 50$K$ images is about 5 hours on a single 3090 24G GPU, but it takes about several days to generate 50$K$ images with diffusion models (about 4 days for MDT [1] and 2.5 days for DG-Diffusion [2]).
> > Consequently, improving both the speed of our evaluation and the inference speed of diffusion models is also important.
> > In the future, we plan to integrate various accelerate techniques to improve our evaluation speed without compromising the evaluation accuracy, such as optimizing the model architecture, model pruning and distillation, *etc*.
> >
> > **Q3. The summary sentences of Section 2.3, 3.1, and 3.2 are not emphasized and paragraph are not strictly parallel.**
> >
> > **Reply:**
> > Thanks.
> > Our presentation is organized for following reasons:
> > In Section 2.3, we present the details of generative models, evaluated datasets, and analysis approaches (including our visualization tool, histogram matching attack, and human evaluation).
> > They are independent of each other, thus we discuss them in parallel in the main paper.
> > In Section 3.1, we investigate the feature extractors by first identifying their attention on visual semantics, followed by investigating their robustness to the histogram matching attack.
> > Finally, we filter extractors that define similar representation spaces.
> > These studies are gradually deepening, thus they are organized in a progressive manner.
> > In Section 3.2, we first study the numerical scales of CKA and FID across various extractors and hierarchical layers of one certain extractor.
> > After that, we investigate the sample efficiency of CKA and KID.
> > In the last paragraph of Section 3.2, we summarize our findings about the feature extractors and the distributional distances.
> > Moreover, the summary sentences of each paragraph provide our primary findings of this paragraph.
> > Following your valuable suggestions, we will carefully proofread and revise the corresponding presentation to make our paper more logical.
> >
> > Hope that the above discussions could address your concerns, please let us know if you have any further questions. Thanks for your effort and constructive suggestions again.
> >
> > [1] Gao, Shanghua and Zhou, Pan and Cheng, Ming-Ming and Yan, Shuicheng. 2023. Masked diffusion transformer is a strong image synthesizer. *ICCV*.
> >
> > [2] Kim, Dongjun and Kim, Yeongmin and Kang, Wanmo and Moon, Il-Chul. 2023. Refining generative process with discriminator guidance in score-based diffusion models. *ICML*.

---

> ### Author Response · Authors · 2023-08-26
> **Response to Reviewer 95Dj**
>
> Dear Reviewer 95Dj,
>
> Thank you so much for your effort in reviewing this submission! We have tried our best to address the mentioned concerns/problems in the rebuttal. Please let us know if you have any additional questions, we are happy to clarify them, thanks for your time and effort again.
>
> Best, Authors.

---

### Official Review · Reviewer_Z9sB · 2023-07-22
**A comprehensive study of evaluation of image synthesis**

**Rating:** 7
**Confidence:** 4

**Strengths:**

- The topic of the paper (evaluation of generative models) is of high importance as the field of generative modeling constantly grows.
- Many generative models are tested in this paper, many feature extractors are employed for the metrics.
- An extensive human evaluation is conducted to compare the metric results against.
- Better sample efficiency of the proposed CKA-based approach (compared to Freschet distance) is a strong advantage of the new metric

**Additional Feedback:**

**I have several questions to the authors.**
- Could you please visualise the tables for better readability?
- Could you please provide a more exanatory discussion of what is CKA (beside the formulas) and some intuition what it measure and why it is a good metric?
- Is CKA a discriminative metric in the sence that CKA(X,Y)=1 if and only if the sets coincide? In other words, what is the "null space" of this metric?
- What is the computational complexity of CKA compared to FID? Could you please explain both theoretical aspect (e.g., O(D^3*N) or so), how it scales with the feature dimension and sample size) and practical aspects (wall clock time)?
- What is the theoretical sample complexity of CKA? Are there any known results here?
- Is centered kernel alignment somehow related to the kernel maximum mean discrepancy (KID/MMD)? A discussion here would be appreciated.

**Clarity:**

The overall clarity is ok. Howver, the explanation of the used Centered kernel alignment metric is very brief and barely understandable. For example, the is no "kernel" in formulas (2) and (3) but in Appendix the authors write that they use exponential kernel and select its bandwith. In fact, the proposed metric seems to have a parameter to be carefully tunes (kernel/its bandwidth), but this limitation is never highlighted in the main text. This feels a little bit unfair w.r.t. the reader. I think the authors should better explain CKA metric in the main text.

**Correctness:**

As I already noted, the table in the paper contain only mean values but nt stds. While in most cases this is not a serious problem (the overall trends are anyway clear), in some cases this is a notable issue. For example, it seems like in Table 2 the authors judge about the reliability of this or that extractor based on a **single restart** (?)  with the histogram matching attack. In almost all the cases, the attack provides only a super **minor** improvement of FID. It is so small that it is unclear whether the result is significant or it is just within the variance of attempts (???). Still based solely on this the aithors judge about the reliability of feature extractors. I am not sure this experimental setup is correct.

**Documentation:**

It seems like this paper does not need a large documentation as it is not a paper which proposed a benchmark or dataset. Still the paper proposes a metric, hence I would expect the authors to clearly describe it and provide a code/recommendation about its evaluation. I did not find the code in the supplementary material.

**Ethics:**

-

**Limitations:**

May be one limitation is that there are no stds reported (only means) and it seems like some of the results are based on a single observation.

**Opportunities For Improvement:**

- In all the cases, only the mean values of metrics are reported, no stds.
- The authors provide many tables with the results but it is not trivial to parse them. Specifically, checking whether this or that metric correlates with the human evaluation ahowld be done manually. It would be great if this could be somehow quantified or visualized (e.g., FID/other metrics as functions of the user score, 2D plots).
- One important advantage of the proposed CKA-based metric compared to FID is the sample efficiency. However, there are other FID-like metrics which are also sample efficient, e.g., Kernel Inception distance (MMD in the feature space) and I regularly meet it alongside FID in generative modeling papers. I would expect the authors to include at least some evaluation/comparison/comment with this metric (both in terms of correlation with human evaluation and sample efficiency).
- On average, I do not see that FID performs notably worse/better than CKA alternatives in correlation with the human evaluation. May be it is more fair to emphasize other advantages of CKA (such as the sample efficiency) rather than consistency and reliability. In the current form, i would argue that the paper overclaims a little bit.

**Relation To Prior Work:**

In general is ok, but since CKA uses "kernels" analogously to KID (MMD), I think it would be nice to devote a paragraph-or-two in the main text about its relation to MMD/KID (if any -- I am not sure it exists).

**Summary And Contributions:**

The most popular metric which is used nowadays to evaluate generative models is FID. It consists of two components: the feature extractor (inceptionV3) and the distributional distance (Freschet distance). The current paper studies potential alternatives to both these components and whether changing these components could somehow benefit the practitioner, e.g., provide a better (in various senses) metric than FID.

To begin with, the paper studies the first component, i.e., feature extractor. It tests various extractors (CNN/ViT based) beside Inception and based on the so-called histogram matching attack desides which are reliable which are not .

The major part of the paper is the evaluation of many existing SOTA or nearly SOTA generative models via using these feature extractors and two variants of the distributional distance - Freschet distance and the CKA metric which the authors propose to use as a substitute to Freschet. In order to have a notion of the ground truth score, the authors do a large user study for all the models in view. Based on the correlation of the tested metrics of the user study, the authors assess the reliability of the tested metrics.

Throughout the evaluation section, the authors devote a lot of time to illustrating that CKA-based metric has many advantages compared to FID such as being normalized, sample efficient. Moreover, it is claimed that this metric is sometimes better correlated with the results of the human evaluation than FID. This means that this metric has a good potential to be a substitute to FID.

UPDATE: 6 to 7. See my response

---

> ### Author Response · Authors · 2023-08-21
> **Response to Reviewer Z9sB (Part1)**
>
> Thanks for the valuable comments. Individual concerns are addressed as follows. Note that, to make sure the revision is easy to track, we do not change the paper structure, and instead list all additional results in the rebuttal of the supplementary material. We will rearrange these materials in the next version. We appreciate the constructive feedback, which has helped to improve the clarity and rigor of our work.
>
> **Q1.Only the mean values of metrics are reported, no stds.**
>
> **Reply:** Thanks.
> As suggested, we add the std values of our experiments to better illustrate the numerical fluctuation of various extractors towards the histogram attack. The following table presents the quantitative results.
> We could tell that the FD scores of extractors that are vulnerable to the attack can be improved by matching the histogram, and the improvement of FD scores is greater than stds.
> For instance, the improvement of FD scores from the Inception model is 0.16 and the computation std is only 0.01, there is an order of magnitude difference between them.
> Moreover, the improvement of FD scores from the Swin-Transformer model is 2.86 and the computation std is only 0.12.
> That is, the improvement is actually caused by the histogram attack rather than the variance of attempts.
> Note that the generator is unchanged but the FD scores are improved by the attack, which is unacceptable for synthesis evaluation.
> Accordingly, extractors that are vulnerable to the histogram matching attack are not reliable for evaluation.
>
> | Model      | Inception        | ConvNeXt          | SWAV              | MoCo$_r$          | RepVGG             | CLIP$_r$          | Swin               | ViT               | DeiT               | CLIP$_v$         | MoCo$_v$         | ResMLP            |
> |------------|------------------|-------------------|-------------------|-------------------|--------------------|-------------------|--------------------|-------------------|--------------------|------------------|------------------|-------------------|
> | Random     | 2.81$\pm$0.01    | 78.03$\pm$0.10    | 0.13$\pm$0.002 | 0.24$\pm$0.003 | 129.61$\pm$0.41 | 10.34$\pm$0.06 | 142.87$\pm$0.12 | 15.11$\pm$0.09 | 437.80$\pm$0.14 | 1.06$\pm$0.01 | 7.32$\pm$0.03 | 99.11$\pm$0.06 |
> | Chosen$_I$ | 2.65$\pm$0.01    | 78.19$\pm$0.11    | 0.13$\pm$0.002 | 0.24$\pm$0.003 | 129.67$\pm$0.39 | 10.36$\pm$0.08 | 140.01$\pm$0.12  | 15.11$\pm$0.10 | 430.81$\pm$0.16  | 1.06$\pm$0.01 | 7.40$\pm$0.03 | 95.36$\pm$0.06  |
>
> **Q2. Visualize the tables to make them more trivial to parse.**
>
> **Reply:** Thanks.
> Following your valuable suggestion, we visualize the correlation between different metrics and the human evaluation results.
> Specifically, we plot the correlation of the averaged ranks of various models given by human judgment, CKA, and FID.
> Fig.2 and Fig.3 in the rebuttal of the *Supplementary Material* respectively present the visualization results of the ImageNet, FFHQ, and LSUN-Church datasets.
> Obviously, the averaged ranks given by CKA are more consistent with that of the human evaluation, demonstrating the accuracy of CKA.
> Moreover, we plot the comparison between the stds and the improvements obtained by the histogram attack for better illustration.
> Fig.4 in the rebuttal of the *Supplementary Material*  presents the results.
> Similarly, we could observe that the improvement is actually caused by the histogram attack rather than the variance of attempts.
>
> **Q3. May be it is more fair to emphasize other advantages of CKA (such as the sample efficiency) rather than consistency and reliability.**
>
> **Reply:**
> Thanks.
> On one hand, our results demonstrate that CKA provides a consistent ranking with the FID scores  in most cases, demonstrating that CKA can deliver the similarity between different data distributions.
> One the other hand, CKA agrees with human visual judgment whereas FID fails in some circumstances.
> That is, CKA can measure the synthesis performance more reliable than FID.
> Additionally, CKA shows better sample efficiency than both FID and KID.
> Thus we integrate CKA as the distributional distance to evaluate the synthesis performance in our system.
> Together with several robust feature extractors, our new measurement system is more consistent and reliable than exiting alternatives.
> In the main paper, we emphasize the reliability and consistency of our entire system rather than only the distributional distance (*i.e.*, CKA) as both the extractors and distances are important.
> We will proofread our presentation and emphasize the advantages of our overall system following your valuable suggestions.

---

> > ### Author Response · Authors · 2023-08-21
> > **Response to Reviewer Z9sB (Part2)**
> >
> > **Q4. Include some comparison with KID.**
> >
> > **Reply:**
> > Thanks.
> > Following your valuable suggestion, we further involve Kernel Inception Distance (KID), precision, and recall into our comparison.
> > Note that the original KID employs Inception-V3 as the feature extractor, and there is a large ``perceptual null space" in Inception-V3.
> > Therefore, we first investigate whether KID scores can be altered by attacking the feature extractor with the histogram matching mechanism.
> > The experimental details are consistent with computing Frechet Distance in Tab.2 of the main paper.
> > The following table presents the quantitative results.
> > Still, some extractors, such as Inception, Swin-Transformer, and ResMLP, are susceptible to the histogram matching attack.
> > For instance, the KID score of Swin-Transformer is improved by 5.31\% when the chosen set is used.
> > These observations agree with our findings in our main paper, suggesting that certain extractors can be hacked when KID is employed as the distributional distance.
> > Then, we investigate the sample efficiency of KID, Precision, and Recall to probe the impacts of the amount of generated samples.
> > Fig.5 in the rebuttal of the *Supplementary Material*  presents the curves of KID, Precision, and Recall scores computed under different data regimes.
> > Similarly, we could observe that the KID scores can be improved by synthesizing more images.
> > Interestingly, the recall scores decrease as the generated sample size increases whereas the precision is stable.
> > This is caused by the definition of recall: recall measures the proportion of the real distribution that is covered by the synthesized distribution.
> > In practical computation, the denominator increases as the synthesized samples increases, while the numerator (*i.e.*, images from the real distribution) remain unchanged.
> > In this way, the recall scores decrease as the generated sample size increases and vice versa.
> > By contrast, CKA scores are stable under different data regimes (please see Fig. 2 in the main paper).
> > Moreover, CKA can provide reliable synthesis evaluation that agrees with human visual judgment.
> > Accordingly, CKA is a proper choice for building a consistent and reliable measurement system.
> >
> > | Model      | Inception                                               | ConvNeXt          | SWAV             | MoCo$_r$         | RepVGG            | CLIP$_r$         | Swin                                                     | ViT               | DeiT              | CLIP$_v$          | MoCo$_v$          | ResMLP                                                  |
> > |------------|---------------------------------------------------------|-------------------|------------------|------------------|-------------------|------------------|----------------------------------------------------------|-------------------|-------------------|-------------------|-------------------|---------------------------------------------------------|
> > | Random     | 1.88$\pm$0.02                                        | 34.81$\pm$0.11 | 9.61$\pm$0.06 | 5.31$\pm$0.06 | 33.88$\pm$0.29 | 2.85$\pm$0.05 | 21.64$\pm$0.10                                        | 16.74$\pm$0.10 | 18.01$\pm$0.19 | 38.06$\pm$0.20 | 15.41$\pm$0.09 | 4.86$\pm$0.02                                        |
> > | Chosen$_I$ | 1.71$\pm$0.02 | 34.82$\pm$0.10 | 9.61$\pm$0.06 | 5.31$\pm$0.05 | 33.89$\pm$0.27 | 2.85$\pm$0.05 | 20.49$\pm$0.09 | 16.74$\pm$0.12 | 19.39$\pm$0.22 | 38.09$\pm$0.19 | 15.40$\pm$0.07 | 4.70$\pm$0.02 |
> >
> > **Q5. The authors should better explain CKA metric in the main text.**
> >
> > **Reply:**
> > Thanks.
> > Following your valuable suggestion, we have added more details of the CKA metric. The revised description can be found in the in the rebuttal of the *Supplementary Material*, and we will move them in the main text in the next version.
> >
> > **Q6. Provide a more explanatory discussion of what is CKA and why it is a good metric.**
> >
> > **Reply:**
> > Thanks.
> > As a widely used similarity index for measuring the correspondence between representations in neural networks, CKA has been identified to have several advantages: 1) CKA is invariant to orthogonal transformation and isotropic scaling, making is stable under various image transformations; 2) CKA can capture the non-linear correspondence between representations due to its kernel mapping;
> > and 3) CKA can determine the correspondence across different features and with different widths, whereas previous metrics fail [1].
> > Additionally, through extensive experiments, we demonstrate that CKA can provide a accurate evaluation for synthesis comparison and is sample-efficient.
> > Accordingly, CKA is a good metric for delivering the distributional discrepancy.
> >
> > [1] Kornblith, Simon and Norouzi, Mohammad and Lee, Honglak and Hinton, Geoffrey. 2019. Similarity of neural network representations revisited. *ICML*

---

> > > ### Author Response · Authors · 2023-08-21
> > > **Response to Reviewer Z9sB (Part3)**
> > >
> > > **Q7. What is the "null space" of CKA?**
> > >
> > > **Reply:**  Thanks.
> > > In fact, it is the feature extractor that might have a "perceptual null space".
> > > For instance, the Inception model has been identified to have a large "perceptual null space", leading it vulnerable to the histogram matching attack.
> > > Moreover, CKA measures the similarity between different distributions, and $\mathrm{CKA(X,Y)} = 1$ if and only if the two sets coincide.
> > >
> > > **Q8. What is the computational complexity of CKA compared to FID?**
> > >
> > > **Reply:**  Thanks. Assume $N$ samples from the evaluated distributions are used for calculating CKA, the main computational complexity of CKA comes from: 1) centering the kernel matrix with the pre-defined centering matrix with the complexity of $\mathcal{O} (N^2)$; and 2) computing HSIC scores with the complexity of $\mathcal{O} (N^3)$.
> > > Therefore, the overall computational complexity of CKA is $\mathcal{O} (N^3)$.
> > > By contrast, the computational complexity of FID mainly comes from calculating the mean and variance of the sample features ($N \times d$, where $d$ denotes the feature dimension).
> > > The computational complexity is $\mathcal{O} (N \times d)^3$.
> > > The computational complexity of both CKA and FID increases linearly with the cubic power of sample size.
> > > Moreover, as suggested, we provide the clock time of FID and CKA in the following table.
> > > Concretely, we use the full FFHQ dataset (70$K$ images) as the reference distribution and generate 50$K$ images for evaluation, the clock time is tested on a single 3090 24G GPU.
> > > We could tell that CKA takes shorter time than FID when the same amount of samples are calculated.
> > >
> > > | Extractor | Inception | ViT      |
> > > |-----------|-----------|----------|
> > > | FID       | 3426 (s)  | 3630 (s) |
> > > | CKA       | 3225 (s)  | 3328 (s) |
> > >
> > > **Q9.What is the theoretical sample complexity of CKA? Are there any known results here?**
> > >
> > > **Reply:**  Thanks.
> > > To the best of our knowledge, there are no known results of the theoretical sample complexity of CKA.
> > > CKA measures the distributional discrepancies between different distributions with a considerable samples from each distribution.
> > > Accordingly, involving sufficient samples for evaluation ensures more accurate results in practice.
> > > However, through evaluating the CKA scores under various data regimes, we observe that CKA shows satisfactory sample-efficiency and stability under different number of samples.
> > > Therefore, we can synthesize subsets with fixed number of images (*e.g.*, 50 $K$) for evaluation.
> > > By contrast, the FID and KID scores could be improved by producing more samples, which is unacceptable for a reliable evaluation.
> > >
> > > **Q10.Is centered kernel alignment somehow related to the kernel maximum mean discrepancy (KID/MMD)?**
> > >
> > > **Reply:**  Thanks.
> > > Centered Kernel Alignment (CKA) is normalized from Hilbert-Schmidt Independence Criterion (HSIC) to ensure it is invariant to isotropic scaling and is formally defined by
> > > \begin{align}
> > >     \mathrm{CKA(X,Y)}=\frac{\mathrm{HSIC}(x,y)}{\sqrt{\mathrm{HSIC}(x,x) \mathrm{HSIC}(y,y)}}.
> > > \end{align}
> > > HSIC is equivalent to maximum mean discrepancy (MMD) between the joint distribution and the product of the marginal distributions, and HSIC with a specific kernel family is equivalent
> > > to distance covariance [1].
> > > HSIC determines whether two distributions are independent, and HSIC = 0 implies independence.
> > > However, HSIC is not invariant to isotropic scaling, making it sensitive to isotropic transformation of images when used for synthesis evaluation.
> > >
> > > Hope that the above discussions could address your concerns, please let us know if you have any further questions. Thanks for your effort and constructive suggestions again.
> > >
> > > [1] Sejdinovic, Dino and Sriperumbudur, Bharath and Gretton, Arthur and Fukumizu, Kenji. 2013. Equivalence of distance-based and RKHS-based statistics in hypothesis testing. *The annals of statistics*.

---

> ### Author Response · Authors · 2023-08-26
> **Response to Reviewer Z9sB**
>
> Dear Reviewer Z9sB,
>
> Thank you so much for your effort in reviewing this submission! We have tried our best to address the mentioned concerns/problems in the rebuttal. Please let us know if you have any additional questions, we are happy to clarify them, thanks for your time and effort again.
>
> Best, Authors.

---

> > ### Comment · Reviewer_Z9sB · 2023-08-26
> > **Results**
> >
> > I have looked through the authors' responses and I see that the authors have adressed and clarified all of my comments. This is an interesting study proposing a promising metric which seems to be better correlated with human judgement than FID. At the same time, it has better sample efficiency. **I increase my score by 1 (6 to 7)**. It would be great if the authors upload the final revision also to take a look how all the changes/new experiments will be incorporated to the text. In this case, I will consider increasing my score further.

---

> > > ### Author Response · Authors · 2023-08-26
> > > **Response to Reviewer Z9sB**
> > >
> > > Dear Reviewer Z9sB,
> > >
> > > Thanks a lot  for your valuable feedback. We are actively preparing the revised manuscript, which still requires some time as we are trying to reorganize the overall presentation and the newly added experiments to better convey the contributions and significance of our manuscript. We will try our best to update the revised version as soon as possible.  We appreciate your constructive suggestions again, which have greatly helped us to improve the quality of our manuscript.
> > >
> > > Best, Authors.

---

> > > ### Author Response · Authors · 2023-08-27
> > > **Response to Reviewer Z9sB**
> > >
> > > Dear Reviewer Z9sB,
> > >
> > > We add this response to clarify how we are preparing the revised version of our manuscript.
> > > The following revisions are incorporated into the main paper:
> > >
> > > +  In the preliminary section, more details of the CKA metric, including the advantages, the kernel section, and the relationship with other metrics such as MMD/KID are added.
> > >
> > > + In the experimental sections, the std results are added as suggested. Due to the limited space of the main paper, we plan to put the visualized figures (The correlation of the averaged ranks of various models given by human judgment, CKA, and FID) in the supplementary, and mention them in the main text. Moreover, for the newly added experiments, including the comparison with other metrics (KID, Precision, and Recall), the comparion on the computational  complexity, and some other experiments suggested by other reviewers, we are still trying our best to organize them to make our manuscript self-contained and easy to follow.
> > >
> > > + We are also proofreading the overall presentation and claims of our manuscript to ensure that our findings and contributions are trivial to parse.
> > >
> > > Due to the limited time, we are still working on this to better convey the contributions and significance of our proposed method.  Again, we appreciate your constructive suggestions, and we are also inspired that you found our responses have adressed your concerns.
> > >
> > > Best, Authors.

---

### Official Review · Reviewer_cWrw · 2023-07-24
**It can solve the current evaluation bias in FID, which takes the average of evaluation results of different architectures and measurements.**

**Rating:** 6
**Confidence:** 2
**Clarity:** Yes.

**Strengths:**

**Strengths:**

1. Comprehensive Investigation: The paper rigorously explores the consistency and comprehensiveness of evaluation paradigms for generative models, addressing the limitations of popular metrics like Fréchet Inception Distance (FID).

2. Feature Extractor Analysis: The study examines the impact of feature extractors on synthesis evaluation, revealing that commonly used extractors, like Inception-V3, may not accurately reflect synthesis improvements due to limited semantics.

3. Incorporation of Multiple Extractors: By incorporating various feature extractors, the research demonstrates potential generalization across domains, leading to a more comprehensive and reliable measurement system.

4. Centered Kernel Alignment (CKA) Metric: Introducing CKA as an evaluation metric provides a more accurate and bounded comparison across different representation spaces.

5. User Study Validation: The extensive user study involving 100 participants confirms that CKA aligns well with human judgment, outperforming FID in some cases, making it a more trustworthy metric.

6. Robust Measurement System: The newly developed measurement system demonstrates its robustness and superiority over FID in re-evaluating state-of-the-art generative models, ensuring consistent and reliable assessments.


**Additional Feedback:**

None.

**Correctness:**

It is a benchmark that takes the average of evaluation results of different architectures and measurements.

**Documentation:**

Yes.

**Ethics:**

No, there are no ethical concerns

**Limitations:**

**Limitations:**

1. Narrow Focus: The paper concentrates mainly on Generative Adversarial Networks (GANs) and may not consider other types of generative models, limiting its applicability to a broader range of models.

2. limited Human Judgment: While the study includes a user evaluation, human judgment can be subjective and I think 100 Human Judgment may not enough to fully capture the complexities of evaluating generative models objectively.


**Opportunities For Improvement:**

1. include the state of the art generative models like diffusion models
2. Since ViT and CNN are compared, it would be interesting if the author can provide the evaluation on MLP based model like map-mixer and see which part does mlp models focus.

**Relation To Prior Work:**

Yes.

**Summary And Contributions:**

**Summary:** The paper investigates image synthesis evaluation using GANs as generative models. It explores data representation, distance calculation, and sample size. Key findings include reliable feature extractors (CNN and ViT), the effectiveness of Centered Kernel Alignment (CKA) for comparison, and CKA's sample efficiency and alignment with human judgment. These findings contribute to a new measurement system for consistent evaluation of generative models.

**Contributions:**
1. Empirical investigation of image synthesis evaluation with GANs.
2. Identification of reliable feature extractors: CNN and ViT.
3. Introduction of Centered Kernel Alignment (CKA) for better comparison.
4. CKA's sample efficiency and alignment with human judgment.
5. Proposal of a new measurement system for consistent evaluation.

---

> ### Author Response · Authors · 2023-08-21
> **Response to Reviewer cWrw (Part1)**
>
> Thanks for the valuable comments. Individual concerns are addressed as follows. Note that, to make sure the revision is easy to track, we do not change the paper structure, and instead list all additional results in the rebuttal of the supplementary material. We will rearrange these materials in the next version. We appreciate the constructive feedback, which has helped to improve the clarity and rigor of our work.
>
> **Q1.Include SOTA generative models like diffusion models.**
>
> **Reply:** Thanks. Following your valuable suggestion, we further include more state-of-the-art generative models on the ImageNet dataset for synthesis evaluation, namely GigaGAN (CVPR'2023) [1], MDT (ICCV'2023) [2], and DG-Diffusion (ICML'2023) [3]. Specifically, we either gather their official models for inference or download the pre-generated images released by the authors for evaluation. Similarly, 50$K$ generated images and the entire training set (*i.e.*, 1.28$M$ images) are used as the synthesized and real distributions, respectively. All details are consistent with the experiments conducted in our main paper. The following table presents the quantitative comparison results. Akin to the results in our main paper, our evaluation system provides consistent ranks with FID and human visual evaluation, demonstrating the reliability of our metric. These results will be added in the next version of our paper.
>
> | Model                               | FID$^\dagger$ | ConvNeXt | RepVGG | SWAV  | ViT   | MoCo-ViT | CLIP-ViT | Overall | User study |
> |-------------------------------------|---------------|----------|--------|-------|-------|----------|----------|---------|------------|
> | GigaGAN [1]              | 3.45          | 68.01    | 79.93  | 90.15 | 98.34 | 82.40    | 96.52    | 85.89   | 65\%       |
> | DG-Diffusion [3] | 3.18          | 68.22    | 80.06  | 90.56 | 98.46 | 82.51    | 96.88    | 86.12   | 66\%       |
> | MDT [2]            | 1.79          | 69.64    | 81.68  | 91.78 | 99.43 | 83.43    | 98.19    | 87.36   | 69\%       |
>
> **Q2. Provide the evaluation on more MLP-based models like mlp-mixer.**
>
> **Reply:** Thanks. As suggested, two MLP-based models are leveraged as the feature extractor for synthesis evaluation, namely gMLP [4] and mixer-MLP[5]. Following the experimental settings in our main paper, we identify the reliability and robustness of these MLP-based models via 1) visualizing the highlighted regions that contribute most significantly to the measurement results, and 2) attacking the feature extractor with histogram matching attack.  The following table presents the quantitative results, please find the qualitative results in the rebuttal of the *Supplementary Material*. On one hand, the heatmap visualization results indicate that both gMLP and mixer-MLP capture limited semantics. Considering that more visual semantics should be considered for a more comprehensive evaluation, gMLP and MLP-mixer might not be adequate for synthesis comparison. On the other hand, the quantitative results demonstrate that their FD scores could be altered by the histogram matching attack, without actually improving the synthesis quality. That is, gMLP and MLP-mixer are susceptible to the histogram attack. Together with the finding that the FD scores of ResMLP could be manipulated without any improvement to the generative models in Tab.2 of our main paper, we do not integrate MLP-based feature extractors into our measurement system. These results will be added in the next version of our paper.
>
> | Extractor | Random             | Chosen$_I$                                                |
> |-----------|--------------------|-----------------------------------------------------------|
> | gMLP [4]      | 2.93$\pm$0.004   | 2.89$\pm$0.004                                             |
> | mixer-MLP [5] | 5.51$\pm$ 0.01 | 5.35$\pm$ 0.01 |
>
> [1] Kang, Minguk and Zhu, Jun-Yan and Zhang, Richard and Park, Jaesik and Shechtman, Eli and Paris, Sylvain and Park, Taesung. 2023. Scaling up gans for text-to-image synthesis. *CVPR*.
>
> [2] Gao, Shanghua and Zhou, Pan and Cheng, Ming-Ming and Yan, Shuicheng. 2023. Masked diffusion transformer is a strong image synthesizer. *ICCV*.
>
> [3]  Kim, Dongjun and Kim, Yeongmin and Kang, Wanmo and Moon, Il-Chul. 2023. Refining generative process with discriminator guidance in score-based diffusion models. *ICML*.
>
> [4] Liu, Hanxiao and Dai, Zihang and So, David and Le, Quoc V. 2021. Pay attention to mlps. *NeurIPS*.
>
> [5] Tolstikhin, Ilya O and Houlsby, Neil and Kolesnikov, Alexander and Beyer, Lucas and Zhai, Xiaohua and Unterthiner, Thomas and others. 2021. Mlp-mixer: An all-mlp architecture for vision. *NeurIPS*

---

> > ### Author Response · Authors · 2023-08-21
> > **Response to Reviewer cWrw (Part2)**
> >
> > **Q3.100 Human Judgment may not enough to fully capture the complexities of evaluating generative models objectively.**
> >
> > **Reply:** Thanks.
> > We agree that involving thousands of persons for human visual evaluation can provide more consistent and reliable results.
> > However, this is too expensive for us as including thousands of participants requires massive human and time resources.
> > Therefore, two strategies of human perceptual judgment are designed for different investigations in our main experiments, namely benchmaking the synthesis quality of one specific generative model and comparing two paired models.
> > In particular, 100 participants are asked to vote the synthesis quality and their final scores are averaged to avoid overly subjective individual outcomes.
> > Moreover, in order to ensure that our human evaluation is reliable and consistent, we repeat the same images several times (*i.e.*, 4) randomly for human visual comparison.
> > In this way, if one user vote photorealistic and unrealistic two times each for the same images, the results would be considered as indistinguishable.
> > This operation further filters overly subjective individual judgment and ensure the rationality of our user study.
> > Additionally, we notice that common choice for human evaluation in the community is to include about 50 participants for perceptual comparison[1, 2, 3, 4, 5].
> > For instance, [1, 2, 3] asked 50 workers to pick the unrealistic images, ProjectedGAN [4] conducted a human preference study with only 28 participants, and the most recent work [5] included only 15 graders to compare the synthesis quality.
> > By contrast, 100 persons are involved in our human judgment, thus we believe that our perceptual comparison results are reliable.
> > Furthermore, we view large-scale human evaluation as our future work to perform more extensive investigations.
> >
> > Hope that the above discussions could address your concerns, please let us know if you have any further questions. Thanks for your effort and constructive suggestions again.
> >
> > [1] Zhu, Jun-Yan and Park, Taesung and Isola, Phillip and Efros, Alexei A. 2017. Unpaired image-to-image translation using cycle-consistent adversarial networks. *ICCV*.
> >
> > [2] Shaham, Tamar Rott and Dekel, Tali and Michaeli, Tomer. 2019. Singan: Learning a generative model from a single natural image. *ICCV*.
> >
> > [3] Kumari, Nupur and Zhang, Richard and Shechtman, Eli and Zhu, Jun-Yan. 2022. Ensembling off-the-shelf models for gan training. *CVPR*
> >
> > [4] Sauer, Axel and Chitta, Kashyap and M{\"u}ller, Jens and Geiger, Andreas. 2021. Projected gans converge faster. *NeurIPS*.
> >
> > [5] Qian, Shengju and Chang, Huiwen and Li, Yuanzhen and Zhang, Zizhao and Jia, Jiaya and Zhang, Han. 2023. StraIT: Non-autoregressive Generation with Stratified Image Transformer. *ArXiv*

---

> ### Author Response · Authors · 2023-08-26
> **Response to Reviewer cWrw**
>
> Dear Reviewer cWrw,
>
> Thank you so much for your effort in reviewing this submission!
> We have tried our best to address the mentioned concerns/problems in the rebuttal. Please let us know if you have any additional questions, we are happy to clarify them, thanks for your time and effort again.
>
> Best, Authors.

---

### Author Response · Authors · 2023-08-21
**Find our point-to-point responses in the revised supplementary material.**

We appreciate all reviewers for their efforts and constructive suggestions. Individual concerns of each reviewer are addressed as follows. Note that, we also provide our detailed point-to-point responses along with additional qualitative and quantitative results in the revised supplementary material (titiled "Rebuttal for Revisiting the Evaluation of Image Synthesis with GANs"), which facilitate the revision process and make the revision easier to track, please find the details via downloading the revised supplementary material. Furthermore, we plan to integrate these additional materials into the revised version of our manuscript and reorganize the overall presentation to better convey the contributions and significance of our proposed method. Furthermore, We will make our code and evaluation scripts publicly available, making it easier to evaluate synthesis performance. We appreciate the reviews' constructive suggestions, which have helped to improve the clarity and rigor of our work, please let us know if you have any further questions.

---

### Decision · Program_Chairs · 2023-09-22

**Decision:**

Accept (Poster)

**Comment:**

The ACs thank the authors and reviewers for their hard work and advancing the frontier in Datasets and Benchmarks. The paper received mostly positive reviews (6, 7, 7, 6, 5). In particular, the reviewers liked the comprehensive study on feature extractors (cWrW, Z9sB), and introduction of CKA (cWrWm 95Dj) which is shown to be superior to FID in several aspects, such as robustness, efficiency, and correlation with user study (cWrW). Furthermore, the authors followed up to the reviewers with extensive evaluation on various settings such as diffusion models (cWrW, y8MJ) and image translation (xEcW). The AC also liked how the paper tackles the evaluation in two distinct aspects of feature extractor and distributional distance metric, performs each analysis comprehensively, and shows clear advantages of the new metric.

There are still a few unaddressed concerns. Most notably, xEcW points out that the new metric is not well studied with respect to the diversity and novelty of the generated samples. Although the authors claim that the CKA, a distributional discrepancy metric, should capture diversity well, the AC agrees with the reviewer that the new metric may ignore the diversity more than FID, as it correlates more with the human preference study, which also ignores diversity. The paper also lacks a separate study on quality vs diversity. There were also additional issues raised by Z9sB and 95Dj on the presentation of the paper, such as insufficient description of CKA and unclear layouts in Section 2 and 3. The author are working to improve the presentation, but it was not ready by the end of the discussion period.

In conclusion, the paper is recommended as a poster presentation.